# FINE-TUNING ENHANCES EXISTING MECHANISMS: A CASE STUDY ON ENTITY TRACKING

**Nikhil Prakash**[1]* **Tamar Rott Shaham**[2] **Tal Haklay**[3] **Yonatan Belinkov**[3] **David Bau**[1]
[1]Northeastern University  [2]MIT CSAIL  [3]Technion – IIT

## ABSTRACT

Fine-tuning on generalized tasks such as instruction following, code generation, and mathematics has been shown to enhance language models' performance on a range of tasks. Nevertheless, explanations of how such fine-tuning influences the internal computations in these models remain elusive. We study how fine-tuning affects the internal mechanisms implemented in language models. As a case study, we explore the property of entity tracking, a crucial facet of language comprehension, where models fine-tuned on mathematics have substantial performance gains. We identify the mechanism that enables entity tracking and show that (i) in both the original model and its fine-tuned versions primarily the same circuit implements entity tracking. In fact, the entity tracking circuit of the original model on the fine-tuned versions performs better than the full original model. (ii) The circuits of all the models implement roughly *the same* functionality: Entity tracking is performed by tracking the position of the correct entity in both the original model and its fine-tuned versions. (iii) Performance boost in the fine-tuned models is primarily attributed to its improved ability to handle the augmented positional information. To uncover these findings, we employ: Patch Patching, DCM, which automatically detects model components responsible for specific semantics, and CMAP, a new approach for patching activations across models to reveal improved mechanisms. Our findings suggest that fine-tuning enhances, rather than fundamentally alters, the mechanistic operation of the model.

## 1 INTRODUCTION

The capabilities of models fine-tuned on general reasoning tasks have hinted at nontrivial mechanisms underlying task learning. While it has been widely understood that fine-tuning a pretrained model on a specific task can improve task performance on that same task (Howard & Ruder, 2018), studies of fine-tuning on generalized domains (Gururangan et al., 2020) have suggested that fine-tuning on generic problems can improve specific task performance as well. In particular, fine-tuning on coding has been observed to lead to a range of improved capabilities in a model (Madaan et al., 2022; Kim & Schuster, 2023). In this paper, we study the mechanisms underlying one specific capability which is dramatically improved by fine-tuning a standard large language model (LLM) on the generic task of arithmetic-problem solving: the ability of a model to perform *in-context entity tracking*, where the model can infer properties associated with an entity previously defined in the input context. For example, if we say "*The apple is in Box C*," a model will later be able to infer "*Box C contains the apple*". The ability to track and maintain information associated with various entities within the context is fundamental for complex reasoning (Karttunen, 1976; Heim, 1983; Nieuwland & Van Berkum, 2006; Kamp et al., 2010), thus making entity tracking an intriguing case study.

We ask several specific questions about the mechanisms underlying the emergence of improved entity tracking in an arithmetic-tuned model. First, we ask: can the performance gap be explained because the fine-tuned models contain a different circuit for performing entity tracking? Or does it contain the same entity-tracking circuit as the base model? To answer this question, we explicitly identify the entity-tracking circuit in the base Llama-7B model, using the path-patching method from Elhage et al. (2021); Wang et al. (2022), consisting of a sparse set of 72 attention heads in four

---

*Correspondence to prakash.nik@northeastern.edu
tamarott@mit.edu, tal.ha@campus.technion.ac.il, belinkov@technion.ac.il, d.bau@northeastern.edu

groups, each group active at a specific token location (Fig. 1); acting in isolation, this sparse circuit can reproduce the entire entity-tracking capability of the base model. Then, without altering the graph, we ask if exactly the same set of components constitutes the entity-tracking circuit in the fine-tuned models. We observe that the identical circuit exists in the fine-tuned models, which alone can restore atleast 88% of the overall performance of the entire fine-tuned model. However, achieving the full performance of the fine-tuned models requires incorporation of additional components.

Next, we ask: how does this common circuit work? Can we discern the role of each group of attention heads? To answer these questions, we use *Desiderata-based Component Masking* (DCM; Davies et al., 2023), a method for automatically identifying model components responsible for performing a specific semantic subtask. That is done by specifying a set of "desiderata," each consisting of pairs of entity tracking tasks, a base task, and a carefully designed alternation of it. The alternation is done on a specific semantic part of the task (*e.g.* the entity name) with a known target output (*e.g.* switch the entity property). Using these sets of tasks, we automatically identify groups of model components that have causal effects that correspond to specific semantics. For example, we could identify whether circuit components are transporting entity name information (*e.g.* "Box C" in the previous example), or its associated property (*e.g.* "contains the apple"), or some other scheme. We test these hypotheses and surprisingly find a third scheme that is used: entity tracking is performed by identifying and transporting the *position* of the queried entity in the context, with multiple groups of heads collaborating to pass the position downstream. Furthermore, this scheme and specific role of each group of heads remain the same between models, confirming that fine-tuning preserves the overall mechanism for performing the entity tracking task. The mechanism invariance is observed in both low-rank adaptations (LoRA) (Hu et al., 2021) and fully fine-tuned models.

Third, we ask: if the mechanism remains the same after fine-tuning, can we attribute the performance improvement to a specific step in the mechanism? To study this question, we introduce *cross-model activation-patching* (CMAP), which allows us to localize the specific sub-mechanism being improved by fine-tuning. Cross-model activation patching shows evidence that (i) the internal representation of both the original model and the fine-tuned models is similar enough so that patching components of the entity-tracking circuit from the fine-tuned models to Llama-7B leads to enhanced performance. (ii) In fine-tuned models the entity tracking circuit has augmented positional information for attending to the correct object and hence fetching its enhanced representation.

Taken together, our findings indicate that fine-tuning enhances the existing mechanism of the original model rather than causing a fundamental shift. Notably, the entity tracking circuit remains consistent across both base and fine-tuned models and maintains the same functionality, with the performance gap mainly attributed to an improved core sub-mechanism. The code, data and fully fine-tuned model can be accessed at `https://finetuning.baulab.info`.

## 2 RELATED WORK

**Mechanistic interpretability** aims to elucidate neural network behaviors by comprehending the underlying algorithms implemented by models (Olah et al., 2017; Elhage et al., 2022). Recently, notable progress has been made in identifying circuits performing various tasks within models (Nanda et al., 2023; Wang et al., 2022; Chughtai et al., 2023; Olah et al., 2020; Lieberum et al., 2023), and in methods enabling circuit discoveries (Davies et al., 2023; Conmy et al., 2024; Wu et al., 2024; Meng et al., 2022; Chan et al., 2022). We aim to harness mechanistic interpretability to uncover an explanation for the performance enhancement observed in fine-tuned models. Specifically, our exploration focuses on whether the performance gap results from varying circuit implementations of the same task and if not, we aim to identify the enhanced mechanism within the circuit.

**Fine-tuning** on generic domains such as code, mathematics, and instructions has been shown to enhance language models performance, both in the context of general fine-tuning and when tailored for specific tasks (Christiano et al., 2017; Gururangan et al., 2020; Madaan et al., 2022; Ouyang et al., 2022; Chung et al., 2022; Taori et al., 2023; Chiang et al., 2023; Liu & Low, 2023; Kim & Schuster, 2023; Zheng et al., 2023; Touvron et al., 2023b; Bommarito II & Katz, 2022). Several attempts to understand the effect of such fine-tuning on model operations reveal interesting characteristics; instruction fine-tuning can destroy knowledge for OOD input (Kumar et al., 2022), shift the model's weight to a task-depended sub-domain (Gueta et al., 2023; Ilharco et al., 2022), and enhance existing capabilities rather than introduce new knowledge (Zhou et al., 2023). Fine-tuned models

were shown to have a localized set of components that perform the task (Panigrahi et al., 2023), and modified underlying embedding spaces and attention patterns (Kovaleva et al., 2019; Merchant et al., 2020; Wu et al., 2020; Zhou & Srikumar, 2022). Concurrent to our research, (Jain et al., 2023) delved into the impact of fine-tuning on LLMs from a mechanistic perspective. Although their main finding, suggesting that fine-tuning rarely alters pretrained capabilities, resonates with our result of enhancing existing mechanisms through fine-tuning, their study involved controlled experiments utilizing transformer models created using the *tracr library* (Lindner et al., 2024). In contrast, our experiments focus on established LLMs such as Llama-7B and their fine-tuned variants, specifically in the context of entity tracking tasks, which we believe better represent real-world language tasks.

**Entity tracking** is a fundamental cognitive ability that enables AI models to recognize and trace entities, including objects, individuals, or concepts, within a given context (Karttunen, 1976; Heim, 1983; Nieuwland & Van Berkum, 2006; Kamp et al., 2010; Marcus, 2018). In the large language models realm, models such as GPT-2 (Radford et al., 2019) have shown some related abilities, such as predicting the next moves in board games (Toshniwal et al., 2022; Li et al., 2022). Utilizing a probing technique, Li et al. (2021) shows that entity state can be recovered from internal activations in BERT (Devlin et al., 2019) and T5 (Raffel et al., 2020). Lately, Kim & Schuster (2023) presented a dataset of entity tracking tasks, showing that models fine-tuned on code data perform entity tracking more accurately. We use entity tracking as a case study to explore how fine-tuning changes the model's functionality to achieve enhanced performance. Complimentary of our work, (Feng & Steinhardt, 2023) investigated how LLMs keep track of various properties associated with an entity. Their findings indicated that models generate binding ID vectors corresponding to entities and attributes. We find it intriguing to further investigate the interaction between these binding ID vectors and the entity tracking circuit we have identified.

## 3 EXPERIMENTAL SETUP

To explore the internal mechanism that enables entity tracking we adapt the dataset presented in Kim & Schuster (2023), aimed at evaluating the ability of a language model to track state changes of discourse entities. The dataset contains English sentences describing different settings of objects located in different boxes, with different labels, and the task is to discover what is inside a specific box. For example, when the model is presented with *"The apple is in box F, the computer is in Box Q, the document is in Box X... Box F contains the"*, it should predict the next token as *"apple"* (see additional task examples in Fig. 2 and in the Appendix J). Each of our tasks involves 7 boxes and no operations (*i.e.* contents of the boxes are not altered), each box is labeled with a random alphabetic letter. For convenience, we only use single-token objects. In contrast to Kim & Schuster (2023), we reorder the structure of the context segment (where each box information is defined) such that the object is mentioned before the box label (*"The apple is in box F"* instead of *"Box F contains the apple"*). This is to ensure that the context segment and the query segment (where the box is queried) have different structures, and the model needs to infer the box information rather than locating the longest identical context segment in the text.

We study four language models: LLaMA-7B (Touvron et al., 2023a), and three fine-tuned versions of it: Vicuna-7B (Chiang et al., 2023) that was fine-tuned on user-shared conversations collected from ShareGPT, Goat-7B (Liu & Low, 2023), fine-tuned on synthetically generated arithmetic expressions using LoRA (Hu et al., 2021), and FLoat-7B (Fine-tuned Llama on arithmetic tasks), fine-tuned on the same data as Goat-7B without LoRA. All these models achieve high performance on the entity tracking task, as shown in Table 1 (first column, evaluation was done over 500 tasks). Although Goat-7B and FLoat-7B were fine-tuned on arithmetic tasks, their ability to perform entity tracking is significantly improved compared to the base Llama-7B model. This aligns with Kim & Schuster (2023), who also found that models trained on structured data are better at performing entity tracking. We seek a mechanistic explanation for this performance gap.

## 4 IS THE SAME CIRCUIT PRESENT AFTER FINE-TUNING?

In this section we ask whether the circuit that enables entity tracking changes across the different fine-tuned models. Entity tracking might be solved by the same circuit in all four models, or each model may implement a different circuit in the light of fine-tuning data. To answer this, we start with

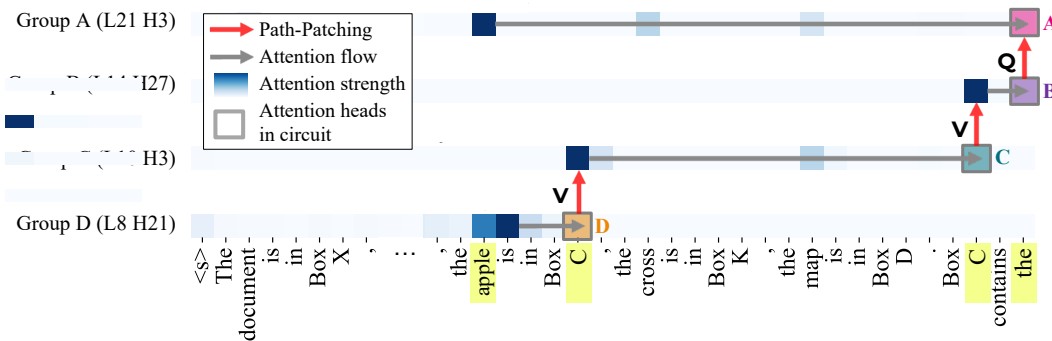

Figure 1: **Entity Tracking circuit in Llama-7B** ($Cir$). The circuit is composed of 4 groups of heads (A,B,C,D) located at the last token (A,B), query label (C), and previous query label (D) token positions. Each group is illustrated by a prominent head in that group.

identifying the entity tracking circuit in Llama-7B, and then evaluate the same circuit components in Vicuna-7B, Goat-7B, and FLoat-7B.

## 4.1 CIRCUIT DISCOVERY IN LLAMA-7B

The entity-tracking circuit will be a subgraph of the transformer computational graph, where each node is an attention head at a specific token position, so the whole circuit is a set $Cir = \{(a, t)\}$. For example, Fig.1 illustrates the entity tracking circuit in Llama-7B consisting of four groups of nodes, each represented by a prominent head; e.g. Group A is characterized with $(a_{\text{L21H3}}, t_{\text{last}})$. Given the nature of the entity tracking task, we are primarily interested in how and what kinds of information are transported between tokens rather than how that information is transformed. We therefore focus our analysis on the attention heads of the circuit, and we consider all MLP layers to be involved in the computation of the final output.

To identify the components of the entity tracking circuit, we use Path Patching (Wang et al., 2022; Goldowsky-Dill et al., 2023), using the synthetic box tracking dataset with 300 examples. For each of the original entity tracking tasks $x_{\text{org}}$ we define a corresponding noise task $x_{\text{noise}}$ with a randomized query, box labels, and objects. Then we evaluate each candidate pair of nodes with a score defined as follows. We denote $p_{\text{org}}$ as the probability of the correct token predicted by the original run, and we let $p_{\text{patch}}$ be the probability assigned to the correct token when patching a specific path from one specific node to another using activations from the noisy run. The patching score for the candidate pair is defined as $(p_{\text{patch}} - p_{\text{org}})/p_{\text{org}}$. At each iteration we add the paths with the lowest (most negative) scores.

In the first step, we identify the group of heads that directly influence the final logit with the lowest patching scores. These attention heads attend mainly to the correct object token: in other words, they look directly at the answer, e.g., 'apple' that should be predicted (Fig. 1). We refer to this set of heads as Group A. We then iteratively identify groups of heads that have high direct effects on each other using the path patching score; this leads us to three additional groups of attention heads, (B, C, and D), active at the last, query label, and previous query label token positions, as shown in Fig. 1. We mark the paths between groups with either Q or V to indicate whether the heads of the previous group affect the query or the value vector calculation of the following group correspondingly.

Overall, the circuit $Cir$ consists of four groups of heads. Group D at the previous query label token collects information of its segment and passes it on to the heads in Group C at the query box label position via V-composition. The output of Group C is transported to the last token residual stream via the heads of Group B through V-composition, which is used by the heads of Group A via Q-composition to attend to the correct object token. The validity of this information flow channel is further substantiated by the results obtained from the attention knockout technique introduced in Geva et al. (2023), as demonstrated in Appendix A. Interestingly, this circuit suggests that correct object information is fetched directly from its token residual stream, instead of getting it from the query label token residual stream. This result is consistent with the findings of Lieberum et al.

Table 1: **Entity-tracking circuit** found in Llama-7B, evaluated on Llama-7B, Vicuna-7B, Goat-7B, and FLoat-7B, *without any adjustment of the circuit graph*. The circuit achieves high accuracy and faithfulness scores in all models (chance accuracy is $0.14$).

| Model | Finetuned? | Accuracy | | | Faithfulness |
|---|---|---|---|---|---|
| | | Full-Model | Circuit | Random Circuit | |
| Llama-7B | – | 0.66 | 0.66 | 0.00 | 1.00 |
| Vicuna-7B | User conversations | 0.67 | 0.65 | 0.00 | 0.97 |
| Goat-7B | Arithmetic tasks (LoRA) | 0.82 | 0.73 | 0.01 | 0.89 |
| FLoat-7B | Arithmetic tasks (w/o LoRA) | 0.82 | 0.72 | 0.01 | 0.88 |

(2023), reporting that heads affecting final logit attend to the correct label, instead of content tokens, to identify the label corresponding to the already-determined correct answer.

## 4.2 CIRCUIT EVALUATION

Although path patching ranks a head based on its relevance via the patching score, it does not provide a clear threshold for the number of heads that should be included in the circuit. In our setting, we include a total of 90 heads in the circuit discovered with path patching (50,10,25,5 heads in Groups A,B,C,D respectively). However, there might be redundancy among the heads in each group. Hence, inspired by Wang et al. (2022), we use a *minimality* criterion to prune the initial circuit. We then measure the performance of the minimal circuit compared with that of the entire model using the *faithfulness* metric. We also evaluate it with the *completeness* metric in the Appendix C.

For both criteria, we define the performance metric $F$ to be the accuracy score averaged over 500 examples. That is, for the model $M$ and its circuit $Cir$, $F(M), F(Cir)$ represent the accuracy of the model and circuit respectively. Specifically, we compute $F(Cir)$ by first mean ablating of all the heads in the model that are not involved in $Cir$.

**Minimality.** The minimality criterion helps identify heads that do not significantly contribute to the circuit performance found with path patching (90 heads in total). For each head, $v \in Cir$, and a subset of heads $K$, we measure the relative performance difference of $Cir$ when the heads in $K$ are knockout, with and without $v$ from the circuit. That is, we define the contribution of each head $v$ to $Cir$ as $(F(Cir \setminus K) - F(Cir \setminus (K \cup \{v\})))/F(Cir \setminus (K \cup \{v\}))$. We filter out the heads with a score lower than 1% (*e.g.* contribute less than $1\%$ to the performance of the circuit in the absence of the functionality defined by subset $K$). Unlike Wang et al. (2022), we use a greedy approach to form the subset for each head in $Cir$ (check Appendix B for more details), and only consider heads that positively contribute to the model performance (*e.g.* contribute to performing of the task). Using this criterion we prune $20\%$ of the heads of the initial circuit, hence reducing the total number of heads to 72 (see Appendix D for exact distribution and heads in each group).

**Faithfulness.** We next measure how good is the identified circuit compared with the entire model. We use the criterion of faithfulness, which is defined as the percentage of model performance that can be recovered with the circuit, *i.e.* $F(Cir)/F(M)$. As shown in Table 1, Llama-7B has a faithfulness score of $1.0$, suggesting identified circuit can recover entire model performance.

## 4.3 CIRCUIT GENERALIZATION ACROSS FINE-TUNED MODELS

As described in section 3, fine-tuned models perform the entity tracking task better than the base Llama-7B. Better performance could be attributed to a superior circuit in the fine-tuned models. Hence, in this subsection, we ask the question of whether the fine-tuned models use a different or the same circuit, *i.e.* with exactly the same group of heads, to perform the entity tracking task.

To answer this, we evaluate the circuit identified in Llama-7B, on the fine-tuned models using the faithfulness criterion. Surprisingly, we find that fine-tuned models have good faithfulness scores for the circuit identified in Llama-7B (without any additional optimization or adaptation) as shown in Table 1. Specifically, Vicuna-7B has almost a perfect faithfulness score of 0.97, while Goat-7B and FLoat-7B exhibit slightly lower scores of 0.89 and 0.88, respectively. As a baseline, we calculate

the average accuracy of 10 random circuits with the same total and per-position number of heads; random circuits have virtually zero accuracy. This suggests that Vicuna-7B utilizes roughly the same circuit as that of Llama-7B to perform entity tracking. Whereas, in Goat-7B and FLoat-7B the same circuit is present, but achieving the complete performance of the fine-tuned models requires the incorporation of additional components.

To further investigate the overlap between the circuits of fine-tuned models and the base model, we identify the entity tracking circuits of the Goat-7B and FLoat-7B models, using the same procedure as in Section 4.1 (Refer to Appendix E and Appendix F). We found that these circuits are significantly larger, consisting of 175 attenton heads and approximately forming a superset of the Llama-7B circuit (Refer to Appendix E4 and Appendix F4 for more details). This finding suggests that fine-tuning is inserting additional components to the circuitry that performs entity tracking.

## 5  IS CIRCUIT FUNCTIONALITY THE SAME AFTER FINE-TUNING?

While the same circuit is primarily responsible for performing entity tracking in both the base and fine-tuned models, the specific functionality of different parts of the circuit remain unknown. In other words, to fully comprehend the underlying mechanism through which these models execute the task, it is crucial to understand the functionalities of the circuit components. There are two hypothesis pertaining to circuit functionality in base and fine-tuned models: (i) The same circuit exists in all four models, but the functionalities it implements may vary, accounting for the performance difference. (ii) The circuits of all models implement the same mechanism, but with an enhanced functionality in fine-tuned models. To investigate these hypotheses, we use the automatic Desiderata-based Component Masking (DCM) method, introduced in Davies et al. (2023), for identifying groups of model components responsible for specific functionalities. First, we use DCM on the groups of heads in the minimal circuit of Llama-7B to identify subsets of heads with specific functionalities, (*e.g.* moving positional information or object values). Then, for each model we apply activation patching on those subsets of heads, to quantify their efficacy on various functionalities.

### 5.1  DESIDERATA-BASED COMPONENT MASKING

The DCM method involves using desiderata for identifying model components responsible for specific functionality. Each desideratum consists of numerous 3-tuple (*original, alternative, target*), where *original* is an original entity tracking task, *alternate* is a carefully designed counterfactual task, and *target* is the desired output, as shown in Fig. 2. If a set of components encodes information regarding the desired semantics, then patching activations from the *alternate* run into the *original* run should alter the model output to *target*. Refer to Davies et al. (2023) for more details.

DCM use gradient descent optimization procedures; For each desideratum, we train a sparse binary mask over potential model components to identify the ones that when patched from counterfactual to original run maximize the target value. Hence, compared to brute-force activation patching, DCM is much more efficient. More importantly, it overcomes a major drawback of activation patching, *i.e.* it can locate the subset of model components that work together to produce the final output.

### 5.2  CIRCUIT FUNCTIONALITY IN LLAMA-7B

To untangle the functionality of groups of heads in the Llama-7B circuit, we define three desiderata, as shown in Fig. 2: (i) *Object* desideratum, which is used to identify model components encoding the value of correct object, (ii) *Label* desideratum, used to identify model components encoding the box label value information, and (iii) *Position* desideratum which can be used to identify model components encoding the positional information of the correct object. Please refer to Fig. 2 caption and Appendix G for additional details about each.

We apply DCM to identify the subset of heads that encode these functionalities in Llama-7B circuit. For each group of heads, we train three binary masks, one for each desideratum, that identify the subset of heads encoding specific functionality (check Appendix H for more details). The results are shown in Table A2. All Group A heads encode the value of correct object in their output. While most of the heads in Group B (71.43%) and C (70.0%) encode positional information of the correct object in their output. The heads of Group D are not profoundly involved in any of the three functionalities.

| Desiderata | Alternate | → Original | Target |
|---|---|---|---|
| (a) Object | The book is in Box A, the cup is in Box B, the computer is in Box C, … Box B contains the ____ | The document is in Box X, the pot is in Box Y, the cross is in Box Z, … Box X contains the ____ | cup |
| (b) Label | The book is in Box A, the cup is in Box B, the computer is in Box C, … Box Y contains the ____ | The document is in Box X, the pot is in Box Y, the cross is in Box Z, … Box X contains the ____ | pot |
| (c) Position | The book is in Box A, the cup is in Box B, the computer is in Box C, … Box C contains the ____ | The document is in Box X, the pot is in Box Y, the cross is in Box Z, … Box X contains the ____ | cross |

Figure 2: **Desiderata for identifying circuit functionality.** We define different desiderata (sets of an original sentence and a carefully designed counterfactual alternation of it with a known target output) to evaluate various hypotheses regarding the functionality of a subset of heads within the circuit. (a) when patching heads encoding information about the correct object, the object information from the alternate run (*e.g.* "cup") is implanted into the original run. (b) heads sensitive to the query box label will be affected by the alternate query label (*e.g.* "Box Y") causing the output to be the object in that box. (c) Position encoding heads represent positional information of the correct object token. Therefore patching these, the model outputs the object at the location of the correct object token in the alternate sequence (*e.g.* "cross" which is located at the same position as "computer".)

We next apply activation patching on this subset of heads, using additional ($N = 500$) samples from the three desiderata, and compute the accuracy with respect to the target value. In order to incorporate randomness in the generated data, we repeated the evaluation ten times with different samples of the test set and report the mean accuracy and standard deviation. The results are shown in Fig. 3, indicating that heads in Group A are primarily responsible for fetching the value information of the correct object. Hence, we refer to this set of heads as *Value Fetcher*. Heads in Group B and C are mainly responsible for detecting and transmitting the positional information of the correct object and are therefore referred to as *Position Detector* and *Position Transmitter*. Since we were unable to establish the functionality of heads in Group D, we used their attention pattern to annotate them. These heads primarily attend to tokens in their own segment, as shown in Fig. 1, hence we refer to them as *Structure Reader* heads.

Overall, the circuit generates correct output by first detecting the positional information of the correct object with Position Detector heads, using the information collected by the Structure Reader heads. The positional information is transmitted to the Value Fetcher heads, by the Position Transmitter heads, which resolves this information to locate the correct object location and fetches its value, to be generated as final output. This indicates that *the model is primarily using positional information* to keep track of in-context entities. Additionally, we have some early evidence that the model is encoding positional information relative to the context segment; see Appendix J for more details.

### 5.3 CIRCUIT FUNCTIONALITY IN FINE-TUNED MODELS

Now that we have identified the functionality of the group of heads in the Llama-7B circuit, we can examine whether this circuit, also present in the fine-tuned models, implements the same or different functionalities across different models. To assess this, we employ activation patching on the same subset of heads of Vicuna-7B, Goat-7B, and FLoat-7B that are involved in a specific functionality.

As shown in Fig. 3, the functionality of the subset of heads remains the same across fine-tuned models. Position Detector and Position Transmitter heads of Vicuna-7B and Goat-7B achieve performance similar to that of Llama-7B, though they demonstrate enhanced accuracy in FLoat-7B. The Value Fetcher heads in fine-tuned models consistently show an improved capability to retrieve the correct object value, *e.g.* Goat-7B can achieve a performance improvement of 20% compared to Llama-7B. Furthermore, we found that both Goat-7B and FLoat-7B circuits implement precisely the same functionality within each group, as depicted in Fig. A8 and Fig. A9. These findings suggest that neither additional functionality nor a shift in functionality is introduced in fine-tuned models. Overall, the results confirm the hypothesis that circuits in fine-tuned models implement the same functionality with the insight that the Value Fetcher in fine-tuned models has a better ability to resolve positional information for fetching the correct object value information.

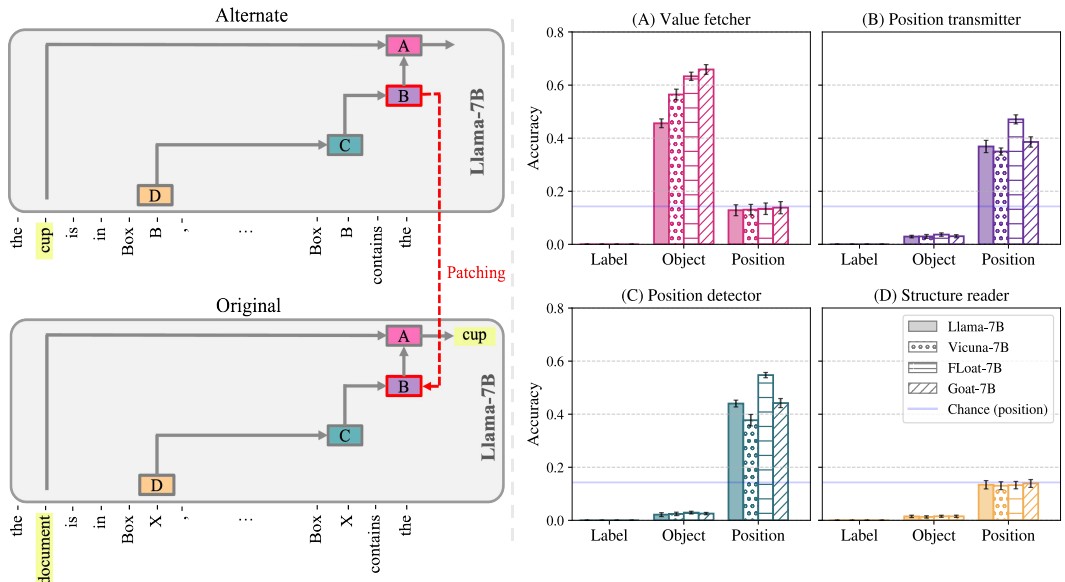

Figure 3: **Circuit Functionality in Llama-7B, Vicuna-7B, Goat-7B, and FLoat-7B.** We use DCM to uncover functionality of each subgroup of Llama-7B circuit. Group A (pink) is mainly sensitive to value desideratum, while groups B, C (purple, turquoise) are responsible for positional information. We find group D insensitive to each of the three desideratum. Error bars indicate standard deviation.

Combining the results from previous experiments indicates that not only the circuit from the base model is present in the fine-tuned models, but also its functionality remains the same. Further, additional components in fine-tuned models' circuits implement the exact same functionality. Hence, we conclude that fine-tuned models implement the same mechanism to perform entity tracking task as the base model. However, the increased performance of fine-tuned models suggests that fine-tuning enhances that existing mechanism. This implies that unraveling the mechanism through which a fine-tuned model accomplishes a task provides valuable insights into how the same task would be executed in the base model. This insight is particularly crucial for tasks that the base model struggles to perform well, making unraveling its mechanism more challenging.

## 6 WHY DO GOAT-7B AND FLOAT-7B PERFORM BETTER?

In the previous sections, we established that fine-tuned models employ the same mechanism as the base model to perform the entity tracking task, albeit with additional components. In this section, we aim to attribute performance improvement to a specific step in the mechanism.

### 6.1 CROSS-MODEL ACTIVATION PATCHING

In order to be able to attribute the performance improvement to a specific step in the mechanism, we introduce *Cross-Model Activation Patching* (CMAP). Unlike naive activation patching, which involves patching activations of the same model on different inputs, CMAP requires patching activations of the same components of *different models* on the same input, as shown in Fig 4.

We use CMAP to patch the output of the subset of heads responsible for dominant functionality in each group of Goat-7B and FLoat-7B circuits. Since we do not fully understand the functionality of Structure Reader heads, we patch all the heads in this group. More specifically, we patch the output of heads in the Goat-7B circuit from the Goat-7B to Llama-7B model, to identify which step in the Goat-7B model mechanism leads to performance improvement. Similarly, we perform the same patching process for heads in the FLoat-7B circuit.

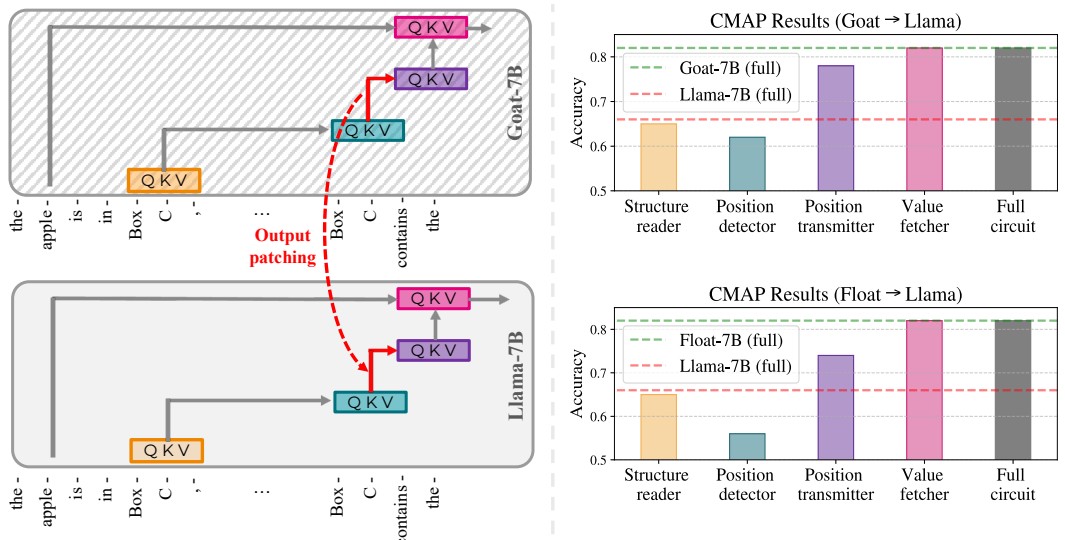

Figure 4: **Why do Goat-7B and FLoat-7B perform better?** We use CMAP to patch activations of the Goat-7B and FLoat-7B circuit components, from Goat-7B and FLoat-7B to Llama-7B model respectively, to attribute the performance improvement to a specific sub-mechanism used to perform entity tracking tasks. We patch the output of the subset of heads in each group that are involved in the primary functionality. We find that patching Value Fetcher heads can solely improve the performance of Llama-7B to that of Goat-7B and FLoat-7B. Additionally, we also observe a significant performance boost when the output of Position Transmitter heads is patched.

## 6.2 RESULTS

As shown in Fig. 4, patching the output of the Position Transmitter and Value Fetcher heads from fine-tuned models to Llama-7B improves the performance of Llama-7B beyond its default performance (red dashed line). It is interesting to observe that the activations of fine-tuned models are compatible with base model, even though they could have been using completely different subspaces and/or norms to encode information. We observe the maximal increase in performance when the Value Fetcher heads are patched, recovering the full fine-tuned models' performance (green dashed line). This indicates that the output of these heads in fine-tuned models encodes an enhanced representation of the correct object, corroborating results from Section 5. Additionally, we also see a substantial increase in performance when the outputs of the Position Transmitter heads are patched, suggesting that fine-tuned models are also transmitting augmented positional information. We speculate that the enhanced encoding in fine-tuned models stem from both additional components in their circuit and the improved ability to encode vital information of shared components with Llama-7B.

## 7 DISCUSSION AND CONCLUSION

In this work, we investigated the effect of fine-tuning on circuit-level mechanisms in LLMs. We discovered that not only does the circuit from the base model persist in the fine-tuned models, but its functionality also remains unchanged. Further, the circuits in fine-tuned models, augmented with additional components, precisely employ the same functionality. We have introduced Cross-Model Activation Patching (CMAP) to compare mechanisms in two different models, revealing how a fine-tuned model enhances the existing mechanism in a base model to obtain a better performance on entity tracking. In our work we have studied the interaction between a single task and three fine-tuned models. Understanding whether such mechanism invariance is typical will require experience with further tasks on more models. Nevertheless, the methods presented in the paper are generic and could be applied to a variety of settings. Future work may study the training dynamics during the fine-tuning process, to pinpoint exactly when and how the circuit enhancement occurs.

## 8 ETHICS STATEMENT

This work investigating the impact of fine-tuning on large language models suggests that fine-tuning primarily enhances existing mechanisms present in the base model. This highlights the importance of training safe and unbiased base models that are openly available. If such models are developed responsibly, then the risks of fine-tuning introducing new biases or dangerous behaviors can be greatly reduced. Hence, indicating that careful stewardship is required in the foundational phases of model development to promote beneficial applications as the capabilities of AI systems advance.

## 9 ACKNOWLEDGEMENT

We would like to thank Open Philanthropy for their generous support through an AI Alignment grant (NP, TH, YB, DB). TH and YB also received support from the Israel Science Foundation (grant No. 448/20) and an Azrieli Foundation Early Career Faculty Fellowship. TRS received partial support from the Zuckerman STEM Leadership Program and the Viterbi Fellowship. We would also like to thank the Center for AI Safety (CAIS) for making computing resources available for this research.

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

## A UNRAVELING CRITICAL INFORMATION FLOW FOR ENTITY TRACKING TASK THROUGH ATTENTION KNOCKOUT

Drawing inspiration from the Attention Knockout technique proposed in Geva et al. (2023), which aims to investigate the flow of crucial information from the subject token to the last token position, we adapted this technique to understand how essential information for entity tracking is conveyed within Llama-7B. In our adaptation, all attention heads of a specific layer and position are obstructed from attending to heads in the same layer at a different position, thereby limiting the flow of information between these positions at the designated layer.

Unlike the approach in Geva et al. (2023), where a window of layers around the specified layer was blocked from attending to a previous position, our method initiates by blocking all layers. Subsequently, at each step, we progressively unblock the next previously blocked layer. More precisely, we block attention heads of all layers at a given position from attending to heads in a different position. Then, in subsequent steps, we systematically unblock each layer, revealing which layer encodes vital information which when unblocked leads to improved performance of the model in conducting entity tracking tasks.

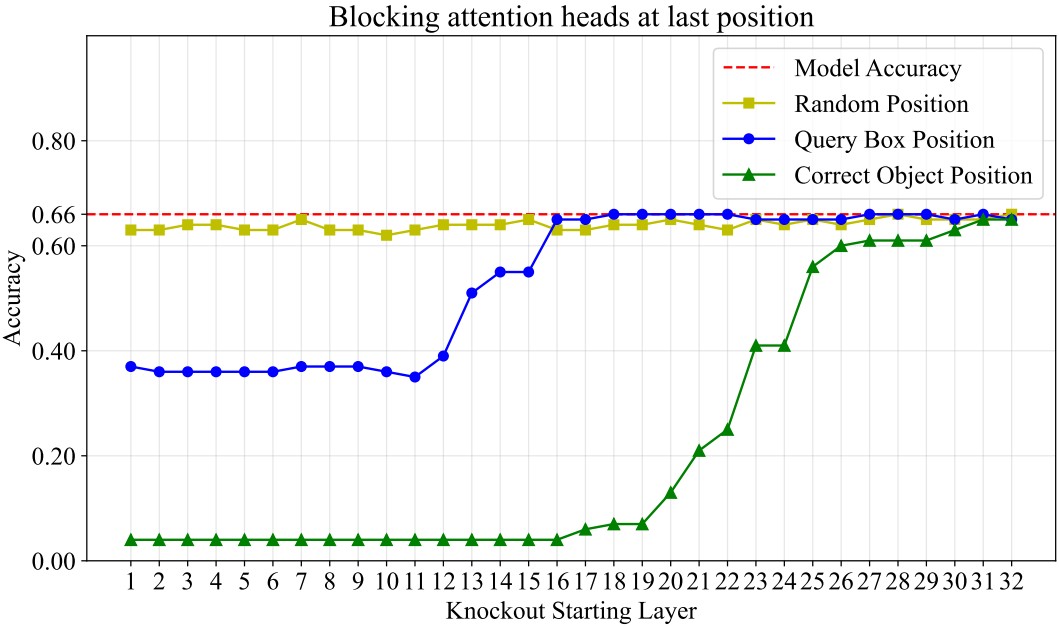

Figure A1: **Critical Information Flow to Last Token Position**

In Fig. A1, the results are presented when heads at the last token position are prevented from attending to various previous positions. It is observed that there are two primary sources of information that heads at the last token position utilize: the query box token and the correct object token position. Specifically, heads in the initial layers focus on and extract crucial information from the query box token residual stream, while heads in the later layers attend to the correct object token, bringing in another essential piece of information. As a baseline comparison, we blocked the heads from attending to a randomly selected previous position, which did not result in a loss of performance. Additionally, Fig. A2 shows the results when heads at the query box token position are blocked from attending to previous positions. We observe that heads in the initial layers transport vital information from the previous query box token position.

The findings from the attention knockout methods align with the information flow subgraph identified using path patching in section. 4.1, as shown in Fig. 1. In other words, heads at the previous query box position collect information about their segment, which is then passed on to the query box token residual stream by the initial layer heads at that position. Subsequently, heads in the initial layers at the final token position attend to the query box token position to incorporate it into their

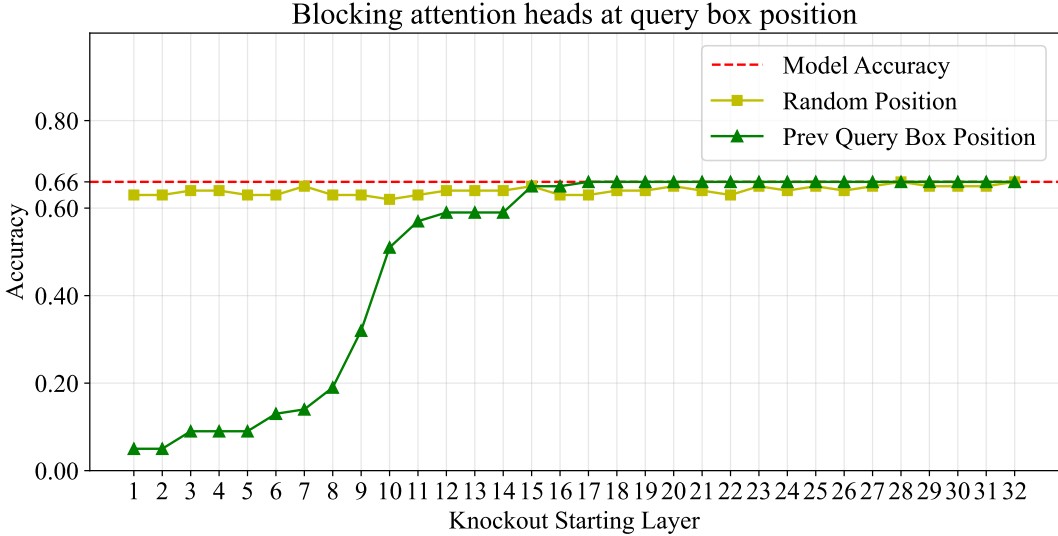

Figure A2: **Critical Information Flow to Query Box Token Position**

residual stream. This information is utilized by the heads in the later layers to attend to the correct object token position and convey it to the final logit.

## B  MINIMALITY

We have utilized the minimality to identify heads that do not contribute significantly to the circuit performance. For each head $v \in Cir$, we seek a subset of heads $K \subseteq Cir \setminus \{v\}$, such that when heads in $K$ are knocked out of $Cir$, $v$ can still recover the performance of $Cir$ considerably. Unlike in Wang et al. (2022) which primarily defined $K$ as the heads in the same class $G \subseteq Cir$ that $v$ belongs to, we use a greedy approach to compute the subset. We cannot use the entire class as $K$, since some of the classes have a large number of heads, e.g. Value Fetcher Heads, that when knocked out would result in a colossal decrease in performance, which cannot be recovered via a single head. To determine the subset $K$ associated with a head $v \in G$, we rank all the other heads in $G$ based on the difference in circuit performance. This difference is computed by comparing the circuit's performance $F(Cir)$, when only the other head $v_j$ (where $v_j$ is any head in $G$ excluding $v$) is removed, and when both $v$ and $v_j$ are removed from the circuit. More specifically, we use $\{F(Cir \setminus \{v_j\}) - F(Cir \setminus \{v, v_j\}) \mid v_j \in G, v_j \neq v\}$ to rank all heads in $G \setminus v$ and then consider upto top 30% of the heads to form the subset $K$.

## C  EVALUATING LLAMA-7B CIRCUIT WITH COMPLETENESS CRITERION

To evaluate the integrity of the Llama-7B entity tracking circuit, we use an additional criterion; *completeness*, which compares the performance of the circuit and the model under knockouts. The circuit is considered complete if eliminating any subset of its components, denoted as $K \subseteq Cir$, results in a comparable performance impact to removing the same subset $K$ from the entire model, as described in Wang et al. (2022). Specifically, for every subset $K$ we calculate the performance of the full model $M$ in the absence of $K$ and compare it to the circuit performance $Cir$ in the absence of $K$, *i.e.* $|F(Cir \setminus K) - F(M \setminus K)|$. We define this as the incompleteness score. If the circuit $Cir$ is complete, it should have a low incompleteness score. Since there are exponentially many $K$, it is computationally intractable to evaluate the incompleteness score of every possible subset $K$. Hence, Wang et al. (2022) proposed a few sampling methods to compute $K$ and we use the following two:

1. **Random**: We uniformly sample a group of circuit components.

2. **Circuit groups**: We define $K$ to be one of four groups of the circuit, *i.e.* A, B, C, and D as shown in Fig. 1.

The results in Fig. A3 and Table A1 indicate that the Llama-7B minimal circuit is not perfectly complete, i.e. there are additional heads in the model that are involved in the entity tracking task. Higher incompleteness score can also be attributed to backup heads, which become active only when other relevant heads are excluded during forward propagation, as identified by Wang et al. (2022).

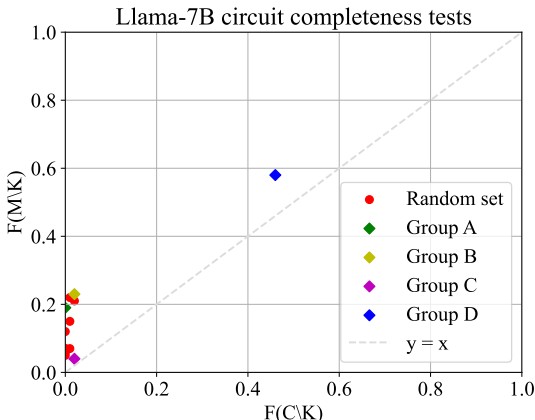

Figure A3: **Completeness of Llama-7B circuit.** We plot the full model accuracy in the absence of a subgroup vs. the circuit accuracy in the absence of the same subgroup. In a complete circuit, this trend should represent equality ($x = y$). Our circuit is not perfectly complete and additional heads might be included.

Table A1: **Completeness evaluation of Llama-7B circuit.** For the random setting we report the mean and standard over 10 random subsets $K$.

|         | Accuracy | | |
| --- | --- | --- | --- |
|         | Full-Model | Circuit | Completeness |
| Random  | $0.11 \pm 0.06$ | $0.0 \pm 0.01$ | $0.1 \pm 0.06$ |
| Group A | 0.19 | 0.0 | 0.19 |
| Group B | 0.23 | 0.02 | 0.21 |
| Group C | 0.04 | 0.02 | 0.02 |
| Group D | 0.58 | 0.46 | 0.12 |

## D   ENUMERATE LLAMA-7B CIRCUIT HEADS IN EACH GROUP

Table A2: **Groups of heads in the Llama-7B circuit.**

| Group | # Heads | Functionality | Name | # DCM |
| --- | --- | --- | --- | --- |
| A | 40 | Value | Value Fetcher | 40 |
| B | 7 | Position | Position Transmitter | 5 |
| C | 20 | Position | Position Detector | 14 |
| D | 5 | - | Structure Reader | - |

**A: Value Fetcher**
L15 H13, L21 H3, L24 H5, L20 H14, L18 H8, L29 H7, L18 H3, L15 H18, L17 H28, L21 H4, L21 H25, L23 H15, L18 H28, L23 H19, L23 H20, L19 H30, L23 H5, L17 H27, L15 H5, L21 H0, L23 H17, L15 H2, L17 H3, L19 H20, L19 H11, L19 H8, L15 H6, L20 H29, L16 H23, L24 H0, L25 H14, L14 H13, L21 H26, L24 H8, L18 H6, L19 H26, L23 H16, L16 H27, L18 H20, L18 H25.
**B: Position Transmitter**

```
L14 H27, L11 H23, L12 H23, L19 H12, L13 H0, L16 H2, L13 H14
```
**C: Position Detector**
```
L10 H3, L13 H14, L9 H2, L9 H7, L11 H23, L9 H10, L1 H9, L7 H17, L13
H0, L6 H10, L4 H4, L7 H26, L9 H21, L8 H1, L12 H0, L8 H22, L10 H4,
L11 H7, L7 H9, L10 H15
```
**D: Structure Reader**
```
L8 H21, L12 H23, L11 H9, L8 H12, L11 H23
```

# E    CIRCUIT DISCOVERY IN GOAT-7B

To better understand the impact of fine-tuning on underlying mechanism for performing entity tracking task, we also identify the circuit in Goat-7B model responsible for performing this task, using the same procedure as described in Section 4.1. The Goat-7B circuit consists of four groups of heads positioned at the same token positions and similar layers, connected through the same type of composition as observed in the Llama-7B circuit. We include a total of 200 attention heads (80,30,50,40 heads in Groups A,B,C,D respectively) in the identified circuit. To eliminate redundant heads, we employ the minimality criterion, mirroring the approach used for the Llama-7B circuit, resulting in a circuit with 175 heads. The distribution of heads among groups is detailed in Table A3. The minimal circuit is evaluated using completeness and faithfulness metrics in the following subsections.

## E1    ENUMERATE GOAT-7B CIRCUIT HEADS IN EACH GROUP

Table A3: **Groups of heads in the Goat-7B circuit.**

| Group | # Heads | Functionality | Name | # DCM |
|-------|---------|---------------|------|-------|
| A | 68 | Value | Value Fetcher | 56 |
| B | 28 | Position | Position Transmitter | 15 |
| C | 40 | Position | Position Detector | 18 |
| D | 39 | - | Structure Reader | - |

**A: Value Fetcher**
```
L24 H5, L17 H28, L20 H14, L21 H4, L21 H3, L18 H8, L23 H19, L19
H30, L30 H4, L23 H17, L29 H7, L19 H20, L23 H15, L31 H6, L28 H16,
L30 H8, L28 H17, L17 H8, L25 H14, L24 H8, L19 H8, L23 H16, L21
H25, L31 H26, L19 H11, L31 H25, L18 H20, L31 H23, L15 H12, L31 H1,
L23 H27, L19 H26, L21 H19, L20 H0, L19 H1, L17 H27, L20 H29, L20
H7, L17 H5, L18 H21, L31 H14, L15 H2, L18 H6, L16 H23, L15 H31,
L19 H23, L21 H11, L23 H31, L31 H0, L17 H24, L11 H5, L22 H17, L13
H10, L14 H9, L18 H23, L15 H24, L21 H17, L16 H27, L19 H2, L17 H23,
L24 H0, L15 H9, L31 H24, L19 H15, L24 H4, L25 H19, L14 H3, L31 H30
```
**B: Position Transmitter**
```
L14 H27, L12 H23, L11 H23, L19 H12, L17 H26, L13 H1, L16 H16, L13
H0, L16 H2, L15 H4, L15 H13, L16 H28, L15 H18, L13 H25, L14 H11,
L14 H13, L10 H6, L13 H27, L12 H25, L12 H8, L12 H0, L11 H9, L18 H3,
L18 H28, L14 H10, L11 H26, L11 H24, L13 H3
```
**C: Position Detector**
```
L10 H3, L13 H14, L6 H10, L11 H23, L11 H24, L9 H7, L1 H9, L9 H10,
L10 H7, L7 H17, L13 H0, L5 H7, L12 H0, L12 H8, L12 H23, L12 H20,
L13 H25, L13 H12, L4 H4, L13 H4, L12 H16, L11 H2, L11 H7, L7 H26,
L10 H4, L4 H27, L9 H21, L8 H1, L11 H28, L12 H30, L8 H12, L9 H30,
L15 H26, L13 H23, L6 H17, L13 H1, L8 H29, L8 H25, L13 H6, L12 H17
```
**D: Structure Reader**
```
L8 H21, L11 H9, L12 H23, L11 H23, L12 H13, L9 H14, L9 H21, L10 H6,
L7 H2, L9 H29, L12 H29, L8 H13, L8 H12, L11 H28, L12 H30, L12 H25,
L8 H10, L8 H26, L8 H7, L10 H24, L6 H23, L9 H30, L11 H24, L9 H18,
L9 H12, L12 H0, L8 H19, L11 H11, L8 H9, L11 H20, L11 H26, L9 H6,
L8 H25, L9 H7, L9 H26, L10 H3, L9 H9, L12 H11, L7 H0
```

## E2   Evaluating Goat-7B Circuit with Completeness Criterion

To assess whether the identified Goat-7B circuit encompasses all the components engaged in executing the entity tracking task within the Goat-7B model, we utilize the completeness metric, as detailed in Section C. We use two sampling methods for computing $K$: 1) Random sampling and 2) Circuit groups.

Fig. A4 and Table A4 presents the completeness score of the Goat-7B circuit, revealing that the circuit is nearly complete, except for heads in Group A. This suggests the possibility of additional heads contributing to the entity tracking task. The higher incompleteness in Group A could also be attributed to backup heads, which become active only when other relevant heads are excluded during forward propagation, as identified by (Wang et al., 2022).

Table A4: **Completeness evaluation of Goat-7B circuit.** For the random setting we report the mean and standard over 20 random subsets $K$.

|  | Accuracy | | |
|---|---|---|---|
|  | Full-Model | Circuit | Completeness |
| Random | $0.13 \pm 0.06$ | $0.16 \pm 0.11$ | $0.06 \pm 0.06$ |
| Group A | 0.41 | 0.21 | 0.2 |
| Group B | 0.13 | 0.11 | 0.02 |
| Group C | 0.06 | 0.11 | 0.05 |
| Group D | 0.62 | 0.63 | 0.01 |

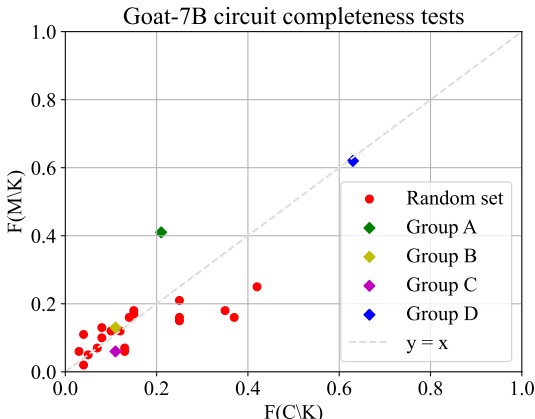

Figure A4: **Completeness of Goat-7B circuit.** We plot the full model accuracy in the absence of a subgroup vs. the circuit accuracy in the absence of the same subgroup. In a complete circuit, this trend should represent equality ($x = y$).

## E3   Evaluation of Goat-7B Circuit with Faithfulness Criterion

For a more thorough evaluation of the Goat-7B circuit in comparison to the entire Goat-7B model, we apply the faithfulness criterion as defined in Section 4.2. As presented in Table A5, Goat-7B attains a faithfulness score of 0.99, indicating that the identified Goat-7B circuit can recover almost the entire model performance. We also observe that the Goat-7B circuit has a similar faithfulness score on FLoat-7B, suggesting a high overlap between the components that perform entity tracking task in these models.

We also observe that the performance of the Goat-7B circuit on Llama-7B and Vicuna-7B is higher than the entire model. Essentially, this implies that when the heads not present in the Goat-7B circuit are mean-ablated in Llama-7B and Vicuna-7B, their performance shows improvement. This phenomenon aligns with observations made in Vig et al. (2020), *i.e.* a small proportion of heads can surpass the gender bias effect of the entire model. Our speculation revolves around the presence of

Table A5: **Entity-tracking circuit found in Goat-7B**, evaluated in Llama-7B, Vicuna-7B, Goat-7B, and FLoat-7B, *without any adjustment of the circuit graph*. The circuit achieves high accuracy and faithfulness scores in all models (chance accuracy is 0.14).

| Model | Finetuned? | Accuracy | | | Faithfulness |
|---|---|---|---|---|---|
| | | Full-Model | Circuit | Random Circuit | |
| Llama-7B | – | 0.66 | 0.77 | 0.00 | 1.17 |
| Vicuna-7B | User conversations | 0.67 | 0.76 | 0.00 | 1.13 |
| Goat-7B | Arithmetic tasks (LoRA) | 0.82 | 0.81 | 0.01 | 0.99 |
| FLoat-7B | Arithmetic tasks (w/o LoRA) | 0.82 | 0.79 | 0.02 | 0.96 |

negative heads, i.e., attention heads that work against predicting the correct object, in these models, contributing to a decrease in their overall performance, as also reported in Wang et al. (2022).

### E4    COMPARISON OF LLAMA-7B AND GOAT-7B CIRCUITS

We observe that, while the faithfulness scores of the Llama-7B and Goat-7B circuits in their respective models are comparable (i.e., 1.0 and 0.99), there is a substantial disparity in their sizes. Specifically, the Goat-7B circuit comprises 175 heads, whereas the Llama-7B circuit has only 72 heads. This notable difference in the number of attention heads in the circuits implies that fine-tuning introduces additional components to the circuitry dedicated to solving the entity tracking task. Indeed, Table A10 and Fig. A5 suggest that the Goat-7B circuit is approximately a superset of the Llama-7B circuit. Additionally, most of the highly causal heads in the Llama-7B circuit remain to be highly influencial in the Goat-7B circuit, suggesting that a minimal alteration in the base model circuit.

Table A6: **Intersection of Attention Heads in Llama-7B and Goat-7B Circuits:** Comparison of the number of heads in each group of Llama-7B and Goat-7B circuits as well as their intersection.

| Head Group | Number of heads in Llama-7B circuit | Number of heads in Goat-7B circuit | Intersection | Precision | Recall |
|---|---|---|---|---|---|
| A | 40 | 68 | 27 | 0.68 | 0.4 |
| B | 7 | 28 | 6 | 0.86 | 0.21 |
| C | 20 | 40 | 16 | 0.8 | 0.4 |
| D | 5 | 39 | 5 | 1.0 | 0.13 |

## F    CIRCUIT DISCOVERY IN FLOAT-7B

In addition to examining the circuits of Llama-7B and Goat-7B, we also identify the circuit in the FLoat-7B model responsible for performing entity tracking tasks. The primary objective of analyzing the FLoat-7B model and its associated circuit is to establish whether the results generalize to models fine-tuned without LoRA. Similar to other circuit identification process, we followed the procedure described in Section 4.1, to discover the Float-7B circuit which also consists of four groups of heads located at the same token positions and similar layers to that in Llama-7B and Goat-7B circuits. We initially included a total of 200 attention heads (80, 30, 50, 40 heads in Groups A, B, C, D, respectively), which were subsequently reduced to 175 heads after applying the minimality criterion. The distribution of heads in the minimal circuit is detailed in Section F1. Notably, we found that the FLoat-7B circuit closely resembles the Goat-7B circuit, suggesting that the impact of fine-tuning with and without LoRA on the underlying mechanism for performing entity tracking tasks is similar. In the following subsections, we evaluate the FLoat-7B circuit using completeness and faithfulness criterion.

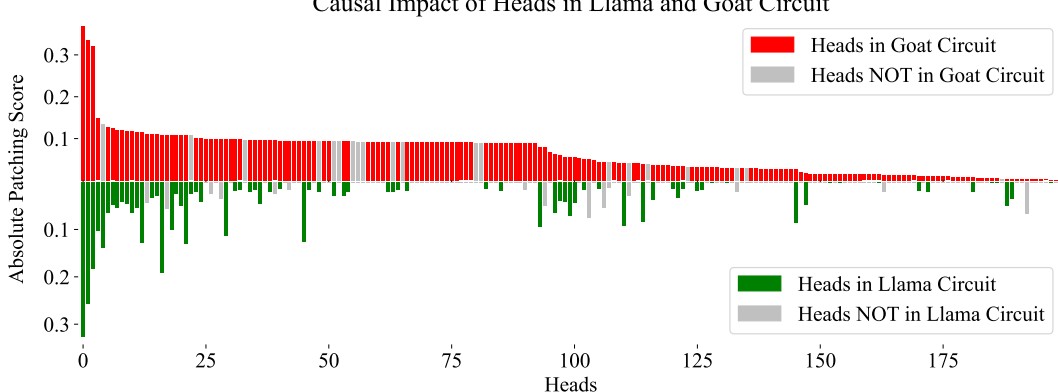

Figure A5: **Causal Impact of Llama and Goat Circuit Heads**: The first subplot shows the attention heads that are included in the Goat circuit (red), sorted based on their causal impact, *i.e.* absolute patching score. For comparison, the second subplot shows the attention heads that are included in the Llama circuit (green) in the same order as the Goat component in the upper subplot (*e.g.* the first bar in both subplots represents the same head that is present in both the circuits, hence colored red and green respectively).

Table A7: **Groups of heads in the FLoat-7B circuit.**

| Group | # Heads | Functionality | Name | # DCM |
|-------|---------|---------------|------|-------|
| A | 68 | Value | Value Fetcher | 60 |
| B | 29 | Position | Position Transmitter | 13 |
| C | 40 | Position | Position Detector | 22 |
| D | 38 | - | Structure Reader | - |

## F1 ENUMERATE FLOAT-7B CIRCUIT HEADS IN EACH GROUP

**A: Value Fetcher**
```
L18 H3, L24 H5, L18 H8, L21 H4, L20 H14, L17 H28, L30 H4, L18 H28,
L23 H17, L19 H30, L23 H19, L28 H16, L15 H5, L19 H20, L31 H26, L15
H18, L25 H14, L19 H11, L15 H12, L29 H7, L17 H3, L30 H8, L19 H1,
L31 H6, L16 H23, L24 H0, L23 H27, L31 H29, L20 H29, L15 H22, L23
H16, L15 H6, L18 H23, L17 H24, L15 H15, L12 H16, L23 H31, L23 H30,
L29 H22, L18 H6, L20 H0, L27 H19, L21 H10, L25 H19, L21 H23, L13
H23, L22 H17, L18 H20, L17 H27, L19 H15, L14 H9, L21 H11, L13 H10,
L17 H5, L14 H13, L20 H26, L18 H25, L19 H24, L18 H21, L24 H11, L12
H9, L31 H23, L26 H16, L16 H15, L19 H17, L17 H15, L19 H16, L22 H5
```
**B: Position Transmitter**
```
L14 H27, L11 H23, L12 H23, L13 H0, L13 H14, L13 H1, L19 H12, L16
H2, L17 H26, L16 H16, L14 H0, L15 H13, L12 H8, L15 H4, L14 H11,
L16 H28, L13 H25, L10 H6, L13 H27, L11 H26, L12 H30, L12 H21, L12
H5, L14 H17, L11 H5, L16 H17, L15 H26, L10 H12, L15 H1
```
**C: Position Detector**
```
L10 H3, L13 H14, L7 H17, L11 H23, L9 H10, L9 H7, L6 H10, L11 H24,
L13 H0, L4 H4, L7 H3, L12 H9, L10 H4, L8 H1, L9 H21, L12 H23, L11
H2, L12 H8, L10 H7, L12 H0, L10 H21, L9 H15, L13 H4, L13 H12, L12
H5, L7 H9, L11 H7, L8 H29, L12 H20, L15 H26, L12 H17, L9 H1, L12
H30, L11 H28, L5 H5, L13 H1, L10 H18, L13 H26, L10 H6, L11 H19
```
**D: Structure Reader**
```
L8 H21, L12 H23, L11 H9, L11 H23, L9 H21, L8 H12, L12 H13, L9 H14,
L8 H11, L10 H6, L11 H28, L9 H30, L12 H29, L6 H17, L9 H12, L8 H7,
L12 H30, L12 H25, L12 H15, L8 H26, L7 H30, L10 H24, L9 H9, L6 H25,
```

```
L8 H13, L11 H24, L5 H4, L9 H26, L12 H11, L10 H12, L9 H6, L11 H11,
L12 H5, L7 H2, L9 H28, L10 H7, L7 H0, L6 H31
```

## F2   EVALUATING FLOAT-7B CIRCUIT WITH COMPLETENESS CRITERION

We assess the entirety for the FLoat-7B circuit using the completeness criterion, as described in Section C. Similar to other circuits' evaluation, we utilize two sampling methods for computing $K$: 1) Random sampling and 2) Circuit groups.

Fig. A6 and Table A8 presents the completeness score of the FLoat-7B circuit. As with Goat-7B circuit, FLaot-7B circuit is almost complete, except for the heads in Group A, suggesting either the presence of additional heads that are fetching the value of correct object or backup heads, that get activated when other relevant heads are ablated during forward propagation.

Table A8: **Completeness evaluation of FLoat-7B circuit.** For the random setting we report the mean and standard over 20 random subsets $K$.

|  | Accuracy | | |
|---|---|---|---|
|  | Full-Model | Circuit | Completeness |
| Random | $0.2 \pm 0.07$ | $0.12 \pm 0.1$ | $0.09 \pm 0.04$ |
| Group A | 0.51 | 0.08 | 0.43 |
| Group B | 0.15 | 0.11 | 0.04 |
| Group C | 0.19 | 0.07 | 0.12 |
| Group D | 0.68 | 0.57 | 0.11 |

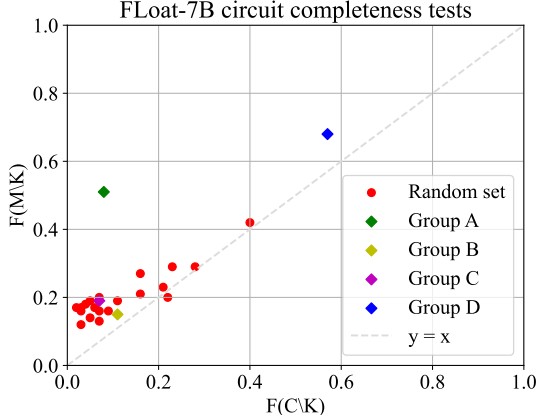

Figure A6: **Completeness of FLoat-7B circuit.** We plot the full model accuracy in the absence of a subgroup vs. the circuit accuracy in the absence of the same subgroup. In a complete circuit, this trend should represent equality $(x = y)$.

## F3   EVALUATING FLOAT-7B CIRCUIT WITH FAITHFULNESS CRITERION

In addition to the completeness score, we assess the FLoat-7B circuit using the faithfulness criterion, which evaluates the performance of the identified circuit compared to the entire model. We observe that the FLoat-7B circuit attains a faithfulness score of $1.0$ on the FLoat-7B model, indicating its capability to recover the entire model performance. Furthermore, we note that its faithfulness on the Goat-7B circuit is $0.93$, suggesting a high degree of overlap between the circuits of the FLoat-7B and Goat-7B models for performing entity tracking.

Table A9: **Entity-tracking circuit found in FLoat-7B**, evaluated in Llama-7B, Vicuna-7B, Goat-7B, and FLoat-7B *without any adjustment of the circuit graph*. The circuit achieves high accuracy and faithfulness scores in all models (chance accuracy is 0.14).

| Model | Finetuned? | Accuracy | | | Faithfulness |
| | | Full-Model | Circuit | Random Circuit | |
|---|---|---|---|---|---|
| Llama-7B | – | 0.66 | 0.69 | 0.0 | 1.05 |
| Vicuna-7B | User conversations | 0.67 | 0.7 | 0.0 | 1.04 |
| Goat-7B | Arithmetic tasks (LoRA) | 0.82 | 0.76 | 0.01 | 0.93 |
| FLoat-7B | Arithmetic tasks (w/o LoRA) | 0.82 | 0.82 | 0.02 | 1.0 |

## F4 COMPARISON OF LLAMA-7B AND FLOAT-7B CIRCUITS

To gain insights into how fine-tuning impacts the underlying mechanism for performing entity tracking in Llama-7B, we compare the identified circuits of Llama-7B and FLoat-7B models. Similar to the observation with the Goat-7B model, we note a substantial difference in the sizes of the Llama-7B and FLoat-7B circuits (72 and 175, respectively). This difference in circuit sizes suggests that fine-tuning introduces additional components dedicated to the task of entity tracking. This is further supported by the results presented in Table A10 and Fig. A7, demonstrating that the FLoat-7B circuit indeed forms a superset of the Llama-7B circuit. Additionally, most of the highly causal heads in the Llama-7B circuit remain highly influential in the Goat-7B circuit, suggesting minimal alteration in the base model circuit.

Table A10: **Intersection of Attention Heads in Llama-7B and FLoat-7B Circuits:** Comparison of the number of heads in each group of Llama-7B and FLoat-7B circuits as well as their intersection.

| Head Group | Number of heads in Llama-7B circuit | Number of heads in FLoat-7B circuit | Intersection | Precision | Recall |
|---|---|---|---|---|---|
| A | 40 | 68 | 27 | 0.68 | 0.4 |
| B | 7 | 29 | 7 | 1.0 | 0.24 |
| C | 20 | 40 | 15 | 0.75 | 0.38 |
| D | 5 | 38 | 5 | 1.0 | 0.13 |

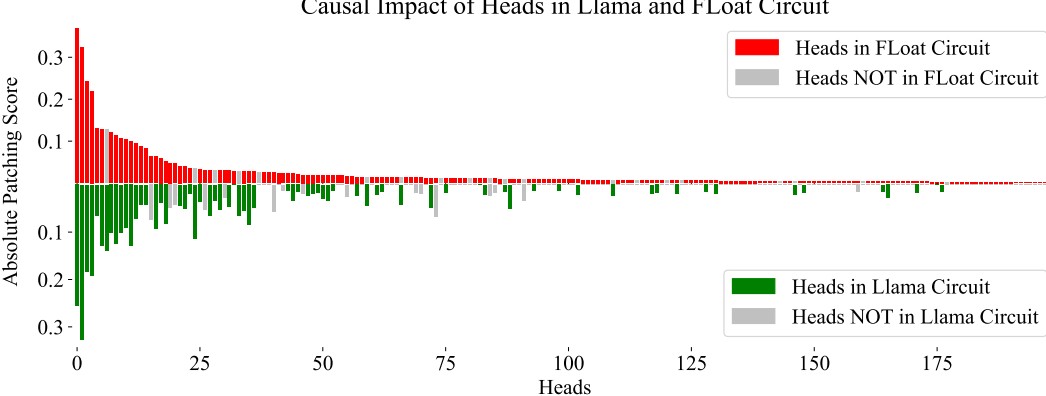

Figure A7: **Causal Impact of Llama and FLoat Circuit Heads**: The first subplot shows the attention heads that are included in the FLoat circuit (red), sorted based on their causal impact, *i.e.* absolute patching score. For comparison, the second subplot shows the attention heads that are included in the Llama circuit (green) in the same order as the FLoat component in the upper subplot (*e.g.* the first bar in both subplots represents the same head that is present in both the circuits, hence colored red and green respectively).

## G DESIDERATA FOR DCM

To identify the functionality of various circuit components, we conceptualized and defined three desiderata:

(1) *Object desideratum* is used to identify attention heads that encode the value of the correct object in their output. Consequently, when the output of these components is patched from the counterfactual run (which contains a different correct object value, associated with a completely different box label) to the original run, the final output changes to the value of the correct object of the counterfactual example, as shown in Fig. 2(a). This occurs even though that object was not included in the original statement.

(2) *Label desideratum* is used to identify circuit components that are encoding the query box label value. Hence, when their output is patched from the counterfactual run (that queried a different box label) to the original run, the final output of the original run changes to the object from the original run that is associated with the query box label of the *counterfactual statement*, as shown in Fig. 2(b). Notably, this occurs even though that object was not initially associated with the query box in the original statement but rather with the query box of the counterfactual one, and it is not present in the counterfactual statement.

(3) *Position desideratum* is used to identify circuit components that encode the positional information of the correct object, i.e. when they are patched from counterfactual run to the original run, the final output of the original run changes to the object in the original statement located at the same position as the correct object of the counterfactual statement, as shown in Fig. 2(c). This happens even though this object is not the correct object of the original run and is not present in the counterfactual statement.

For each of the three desiderata, we train a binary mask over the model components to identify the circuit components encoding the corresponding vital information to accomplish the task.

## H DCM EXPERIMENT DETAILS

As mentioned in section 5.2, we restricted the model component search space to the heads in each group. More specifically, for each group in the circuit, we identify the subset of heads that are involved in three functionalities: 1) encoding object value, 2) box label value, and 3) correct object position. We synthetically generated training ($N = 1000$) and eval datasets ($N = 500$), according to the desiderata. To train a binary mask consisting of learnable parameters ($W$), we minimize the following loss function:

$$\mathcal{L} = -\text{logit}_{\text{target}} + \lambda \sum 1 - W \tag{1}$$

with $\lambda = 0.01$. We trained it for two epochs, with ADAM optimizer and a batch size of 32.

## I CIRCUIT FUNCTIONALITY IN GOAT-7B AND FLOAT-7B

To identify the functionality of the head groups within the Goat-7B and FLoat-7B circuits, we adopted a procedure similar to the one detailed in section 5. Initially, we employed DCM to localize the subset of heads within each group encoding various functionalities. Subsequently, we conducted activation patching on these subsets of heads.

As depicted in Table A3 as well as Fig. A8 for Goat-7B circuit, and in Table A7 as well as Fig. A9 for FLoat-7B circuit, the functionality of each head group mirrors that of the Llama-7B circuit. Specifically, Group A heads predominantly encode the value of the correct object, while Groups B and C are tasked with encoding the positional information of the correct object. Nevertheless, the functionality of heads in Group D continues to remain a mystery.

## J ADDITIONAL DESIDERATA FOR POSITIONAL INFORMATION

In section 5, we found that the models are using the positional information of the correct object for entity tracking. However, we could not precisely characterize the positional information. Therefore,

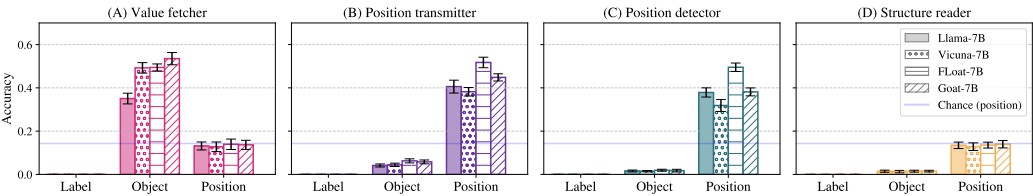

Figure A8: Activation patching results of the functionality heads in the Goat-7B circuit, identified using DCM. Error bars indicate standard deviation.

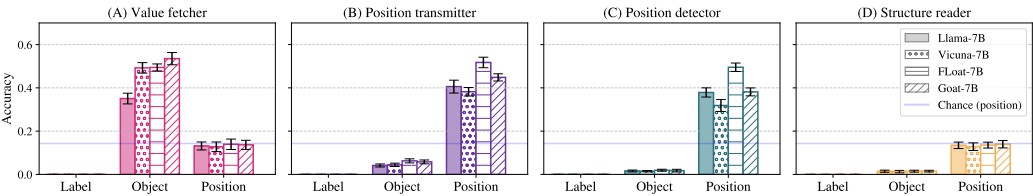

Figure A9: Activation patching results of the functionality heads in the Float-7B circuit, identified using DCM. Error bars indicate standard deviation.

we devised a few additional positional desiderata, as shown in Tables A12 and A13. Similar to our analysis on previously defined desiderata, we applied activation patching on the subset of heads in the Position Transmitter group that encodes positional information. In other words, we were interested in understanding what kind of positional information is being used by Value Fetcher heads to locate the correct object in the context.

Results are summarized in Table A11. Although redundant text at the start, end, or between the object and the box does not impact the positional information encoded in those heads, an additional segment at the start does seem to affect it. Further, mentions of boxes with labels before the correct segment (the segment containing the correct object) also interfere with the positional information. In addition to relative positional information, the semantics of the association between the object and the box are also influential. Combining these results, we speculate that the model is enumerating the association of the boxes and their corresponding object from the start token as well as keeping track of the semantics of the association. However, more work is needed to completely characterize the positional information.

Table A11: **Positional desiderata** to characterize the positional information transported by the Position Transmitter heads to the Value Fetcher heads. Please refer to Table A12 and A13 for examples.

|  | Position hyp. | Acc. | Var. |
|---|---|---|---|
| 1 | Random text at the start | 0.35 | 0.02 |
| 2 | Random text at the end | 0.34 | 0.01 |
| 3 | Additional tokens between object and box | 0.38 | 0.02 |
| 4 | Additional segment at the start | 0.25 | 0.02 |
| 5 | Additional segment at the end | 0.31 | 0.02 |
| 6 | Additional boxes before correct segment | 0.2 | 0.02 |
| 7 | Incorrect box segment | 0.16 | 0.01 |
| 8 | Altered box object order | 0.16 | 0.01 |
| 9 | Altered association btw box and object | 0.18 | 0.02 |
| 10 | No comma to separate segments | 0.38 | 0.02 |
| 11 | Additional comma after the object | 0.36 | 0.02 |

Table A12: **Examples of Positional desiderata**: For each positional desiderata reported in Table A11, this table contains specific examples of original, alternate, and target data.

| Desiderata | Base | Source | Target |
|---|---|---|---|
| 1 | The document is in Box X, the pot is in Box T, the magnet is in Box A , the game is in Box E, the bill is in Box M, the cross is in Box K, the map is in Box D. Box S contains the ____ | There are a bunch of boxes containing objects, the magnet is in Box O, the bell is in Box M, the leaf is in Box W , the cup is in Box G, the ice is in Box J, the milk is in Box Z, the wire is in Box H. Box W contains the ____ | magnet |
| 2 | The document is in Box X, the pot is in Box T, the magnet is in Box A, the game is in Box E, the bill is in Box M, the cross is in Box K, the map is in Box D . Box A contains the ____ | The document is in Box Q, the bus is in Box F, the camera is in Box R, the glass is in Box W, the magazine is in Box Z, the coffee is in Box E, the watch is in Box C , these are a bunch of boxes containing objects. Box C contains the ____ | map |
| 3 | The document is in Box X, the pot is in Box T, the magnet is in Box A, the game is in Box E, the bill is in Box M , the cross is in Box K, the map is in Box D. Box A contains the ____ | The pot is in Box U, the flower is in Box D, the car is in Box K, the disk is in Box C, the fan is contained in the Box H , the bill is in Box S, the painting is in Box L. Box H contains the ____ | bill |
| 4 | The document is in Box X, the pot is in Box T, the magnet is in Box A, the game is in Box E, the bill is in Box M , the cross is in Box K, the map is in Box D. Box A contains the ____ | The apple is in Box O, the dress is in Box N, the boot is in Box Y, the hat is in Box L, the bus is in Box X , the painting is in Box F, the drug is in Box J, the string is in Box D. Box X contains the ____ | game |
| 5 | The document is in Box X, the pot is in Box T, the magnet is in Box A, the game is in Box E, the bill is in Box M, the cross is in Box K, the map is in Box D . Box A contains the ____ | The hat is in Box K, the plane is in Box H, the tie is in Box U, the wire is in Box F, the file is in Box R, the note is in Box Y, the train is in Box G , the apple is in Box O. Box G contains the ____ | map |
| 6 | The document is in Box X, the pot is in Box T, the magnet is in Box A, the game is in Box E , the bill is in Box M, the cross is in Box K, the map is in Box D. Box A contains the ____ | The hat is in Box K, the plane is in Box H, the tie is in Box U, there are three additional boxes, Box PP, Box BB and Box AA, the wire is in Box F , the file is in Box R, the note is in Box Y, the train is in Box G. Box F contains the ____ | game |
| 7 | The document is in Box X, the pot is in Box T, the magnet is in Box A, the game is in Box E, the bill is in Box M , the cross is in Box K, the map is in Box D. Box A contains the ____ | The magnet is in Box O, the bell is in Box M, the leaf is in Box W, the cup is in Box G, the ice is in Box J , the milk is in Box Z, the wire is in Box H. Box J contains the ____ | cross |
| 8 | The document is in Box X, the pot is in Box T, the magnet is in Box A, the game is in Box E , the bill is in Box M, the cross is in Box K, the map is in Box D. Box A contains the ____ | The pot is in Box U, the flower is in Box D, the car is in Box K, Box C contains the disk , the fan is in Box H, the bill is in Box S, the painting is in Box L. Box C contains the ____ | game |

Table A13: **Examples of Positional desiderata**: For each positional desiderata reported in Table A11, this table contains specific examples of original, alternate, and target data.

| Desiderata | Base | Source | Target |
|---|---|---|---|
| 9 | *The document is in Box X, the pot is in Box T, the magnet is in Box A, the game is in Box E, the bill is in Box M, the cross is in Box K , the map is in Box D. Box A contains the ____* | *The ticket is in Box N, the book is in Box J, the gift is in Box W, the coat is in Box Y, the rose is in Box K, the wheel is not in Box G , the brick is in Box V. Box G contains the ____* | cross |
| 10 | *The document is in Box X, the pot is in Box T, the magnet is in Box A, the game is in Box E, the bill is in Box M , the cross is in Box K, the map is in Box D. Box A contains the ____* | *The pot is in Box U the flower is in Box D the car is in Box K the disk is in Box C the fan is in Box H the bill is in Box S the painting is in Box L. Box H contains the ____* | bill |
| 11 | *The document is in Box X, the pot is in Box T, the magnet is in Box A, the game is in Box E , the bill is in Box M, the cross is in Box K, the map is in Box D. Box A contains the ____* | *The clock, is in Box M, the bomb, is in Box J, the newspaper, is in Box G, the letter, is in Box L , the suit, is in Box Y, the computer, is in Box R, the wheel, is in Box V. Box L contains the ____* | bill |

