# OpenReview forum: "Fine-Tuning Enhances Existing Mechanisms: A Case Study on Entity Tracking"
_ICLR.cc/2024/Conference — ICLR 2024 poster_

### Official Review · Reviewer_AnWX · 2023-10-31

**Soundness:** 3 good
**Presentation:** 3 good
**Contribution:** 2 fair
**Rating:** 5
**Confidence:** 4

**Summary:**

This paper examines the impact of fine-tuning on the internal computations of language models through a case study. Specifically, the authors investigate the entity tracking mechanism of Llama-7B and its fine-tuned versions. The authors argue that the performance enhancements resulting from fine-tuning can be attributed to the improved ability of attention heads to handle positional information.
They utilize Desiderata-based Component Masking (DCM) to confirm that the entity tracking mechanism and the functionality of a subset of attention heads remain the same in Llama-7B and its fine-tuned variants. Additionally, they introduce CrossModel Activation Patching (CMAP) to reveal the improved mechanisms of attention heads.

**Strengths:**

1. The paper is generally well-written and easy to follow.

2. The authors provide an explanation for the performance enhancement observed in the fine-tuned model, focusing on the entity tracking circuits.

**Weaknesses:**

1. The methods employed in this paper are limited to a single type of model on an entity tracking dataset. Moreover, the entity tracking mechanism is likely just one of many contributing factors, raising questions about the generalizability of their claims.

2. I personally do not see the significance of the entity tracking problem to be particularly high.

**Questions:**

Why is it justified to employ attention mechanisms to delve into the entity-tracking circuit? I acknowledge there may be some associations between them, but are these connections truly substantial?

---

> ### Author Response · Authors · 2023-11-14
> **Response for Reviewer AnWX**
>
> Thanks for the review! We’re glad that you found that paper clear and easy to follow. We plan to answer your questions with an updated version of the paper soon, but we wanted to address a few of them quickly here.
>
> *Note: Kindly review the manuscript for information regarding references.*
> ***
>
> **The methods employed in this paper are limited to a single type of model on an entity tracking dataset. Moreover, the entity tracking mechanism is likely just one of many contributing factors, raising questions about the generalizability of their claims.**
> - Although we have focused on the specific task of entity tracking, the methods employed and introduced in this work are generic and can be easily applied to other tasks and models, by generating relevant data. Both path patching and DCM have already been applied to other tasks like indirect object identification and variable binding in arithmetic expression (Wang et. al (2022), Davies et. al (2023)). Additionally, CMAP can also be employed among various kinds of models that share the same dimensionality.
> - While we have meticulously analyzed the impact of fine-tuning on the entity tracking mechanism, we acknowledge that understanding whether such mechanism invariance is universal would require further studies involving additional tasks and models. *We have updated our discussion section to clarify this limitation.*
> ***
>
> **I personally do not see the significance of the entity tracking problem to be particularly high.**
> - We understand not all researchers might be interested in the same topics, but there is a significant community of researchers who are interested in both entity tracking and name binding: it has been a significant area of study in not just AI, but also in Linguistics and Cognitive Neuroscience (Karttunen et. al (1976), Heim et. al (1983), Nieuwland et. al (2006), Kamp et. al (2010), Marcus et. al (2018)).
> - For instance, Kamp et. al (2010) proposed multiple structures for discourse representation, with the fundamental atomic steps as binding names with their attributes defined in the context, making it crucial for understanding the discourse capabilities of LLMs.
> - Additionally, Marcus et. al (2018) argued that representing abstractions, instantiating variables with instances, and applying operations to those variables are indispensable to the human mind, which is often ignored in AI research.
> - While we are among the first to study the internal mechanism for the entity tracking task, multiple researchers have previously investigated such capabilities in various deep neural networks, particularly using probing classifiers (Li et. al (2021), Li et. al (2022), Kim et. al (2023)). This interest is current and shared by others; for example, there is another anonymous submission to ICLR (https://openreview.net/forum?id=zb3b6oKO77) that examines the name-binding capabilities from another perspective, indicating the current interest within the research community in this topic.
>  - *We have also updated our manuscript to highlight these points to provide context on the significance of the entity tracking task.*
> ***
>
> **Why is it justified to employ attention mechanisms to delve into the entity-tracking circuit?**
>
> To discover the information flow circuit for the entity tracking task, path patching retraces the crucial components from the end. As some of the heads in the circuit are primarily attending to heads at other tokens, the information they pass on to later components mainly comes from the heads at other tokens, due to the autoregressive structure. Consequently, we use the attention pattern to locate additional components in the circuit.
> ***
>
> **I acknowledge there may be some associations between them, but are these connections truly substantial?**
>
> We are confident that associations between groups of attention heads in the identified circuit are substantial due to multiple pieces of evidence triangulating it.
>  - First, with the path patching algorithm, we observe that these are the paths/connections which when substituted from corrupt run to clean run lead to the most degradation in output probability of correct token.
>  - Second, when computing the faithfulness score of the identified circuit, we mean ablate all the heads except those in the circuit and yet observe a good performance on the entity tracking task, suggesting the paths/connections are indeed causally influential to how the output is generated.
>  - Third, these paths/connections are not only substantial in a single base model (Llama-7B), but also in its fine-tuned versions, as can be concluded from the high faithfulness scores on these models.
>  - Fourth, the cross-model activation patching experiment clearly shows that patching these paths/connections leads to improvement in the performance of the base model.
>  - Finally, we would like to mention that all the techniques (Path patching, DCM, and CMAP) used in this work are causal, rather than correlational or observational, in nature.

---

> > ### Comment · Reviewer_AnWX · 2023-11-22
> > **Response read.**
> >
> > I have read the response to my questions and deciced to keep my rating.

---

### Official Review · Reviewer_fhvV · 2023-10-31

**Soundness:** 3 good
**Presentation:** 3 good
**Contribution:** 3 good
**Rating:** 6
**Confidence:** 3

**Summary:**

This work aims to answer the question: why fine-tuning language models (LMs) can enhance their performance on a range of tasks? As a case study, the authors experiment with LLaMA-7B and two fine-tuned versions, Vicuna-7B and Goat-7B, on the entity tracking task. They first apply Path Patching to extract the circuit, a subset of LM heads, from LLaMA-7B and find that all three models can reach high faithfulness scores with the circuit identified in Llama-7B, a.k.a., all three models share a similar circuit. Then, they apply Desiderata-based Component Masking (DCM) to identify LM heads responsible for specific functionality and find that each group of LM heads on all three models share the same functionality. Finally, they propose the Cross-Model Activation Patching (CMAP) to attribute the performance gain of fine-tuning to specific components, a.k.a., Value Fetcher, and Position Transmitter heads.

The work conducts extensive experiments to uncover the internal mechanisms of fine-tuning by applying two existing methods, Path Patching and DCM, and proposing one novel method, CMAP. The authors disclose some interesting experiment findings, e.g., the performance gain of fine-tuning attributes to the improved ability to handle positional information.

**Strengths:**

- A novel method CMAP for mechanistic interpretability.

- The authors conducted extensive experiments by applying Path Patching, DCM, and their proposed CMAP to analyze the underlying mechanism of fine-tuning, which discloses several exciting findings, e.g., (1) The language model (LM) and its two fine-tuned versions share the same circuit. (2) The components of this circuit in these three models share the same functionality. (3) The performance gain of fine-tuning attributes to the improved ability to handle positional information.

- The presentation is clear, although it requires the reader to have some background of mechanistic interpretability.

**Weaknesses:**

- The experiment results can not lead to the claim that the original model and its fine-tuned versions implement entity tracking with the “same” circuit: (1) the fine-tuned model, Goat-7B, reaches an accuracy of 82% while the circuit identified in Llama-7B reaches an accuracy of 68%; this considerable performance gap indicates that Goat-7B’s circuit may be different from Llama-7B’s circuit, although these two circuits may have considerable overlap. (2) It is necessary to explore the mentioned overlap. For example, could authors apply Path Patching to Vicuna-7B and Goat-7B to extract their circuits and compute the overlap between their circuits and Llama-7B’s circuit?

- As mentioned in (Wang et al., 2022) [1], faithfulness is not sufficient to prescribe which circuits explain the behavior well. Why do not authors show the completeness and minimality scores, similar to (Wang et al., 2022) [1]?

- In Section 4.2: CIRCUIT EVALUATION, the expression F(Cir \ K) - F(Cir \ (K U {v})) / F(Cir \ (K U {v})) in the Minimality paragraph seems wrong. Moreover, the authors mention that they filter out the heads that contribute less than 0.5% of the functionality defined by subset K. This description is inconsistent with the above expression since the denominator of the equation is the functionality defined by the remaining nodes after removing K and v.

[1] Wang, Kevin, et al. "Interpretability in the wild: a circuit for indirect object identification in gpt-2 small." arXiv preprint arXiv:2211.00593 (2022).

**Questions:**

- What is the motivation to divide heads into four groups (A, B, C, D) instead of three or five? In other words, since authors iteratively identify groups of heads with high direct effects on each other using the path patching score, why did authors end with four groups instead of other numbers?

---

> ### Author Response · Authors · 2023-11-22
> **Response for Reviewer fhvV [1/3]**
>
> Thanks for the review! We’re glad that you found the paper clear and the experiments extensive. We have incorporated your suggestions in our revised version, but also wanted to respond to your questions here:
> ***
>
> **The experiment results can not lead to the claim that the original model and its fine-tuned versions implement entity tracking with the “same” circuit: (1) the fine-tuned model, Goat-7B, reaches an accuracy of 82% while the circuit identified in Llama-7B reaches an accuracy of 68%; this considerable performance gap indicates that Goat-7B’s circuit may be different from Llama-7B’s circuit, although these two circuits may have considerable overlap. (2) It is necessary to explore the mentioned overlap. For example, could authors apply Path Patching to Vicuna-7B and Goat-7B to extract their circuits and compute the overlap between their circuits and Llama-7B’s circuit?**
>
> -   We acknowledge that while the performance of the circuit identified in Llama-7B on Goat-7B is satisfactory, its faithfulness score is slightly lower than that of the circuit on Llama-7B, suggesting that the circuit employed by Goat-7B might have a few additional components. Hence, we applied path patching (as suggested) on the Goat-7B model to recover its circuit.
>
> -   We found that similar to the circuit identified in Llama-7B, the Goat-7B circuit comprises of four sets of attention heads, each exhibiting a similar information flow mechanism to that of the Llama-7B circuit. Using the minimality score, we pruned out redundant heads, making it a circuit with 76 attention heads, comparable to the previously identified Llama-7B circuit consisting of 75 heads.
>
> -   We evaluate the performance of the Goat-7B circuit across models and compare its performance to the circuit found in Llama-7B (see the following table):
>
>
> 	1.  Not only the performance of the Goat and Llama circuits in Goat-7B is similar (0.70 vs. 0.68), but their faithfulness scores are also comparable (0.85 vs. 0.83), indicating that the two circuits have roughly the same performance.
>
> 	2.  The identified Goat circuit also performs well on other models. For instance, the Goat circuit could achieve the performance of the entire Vicuna-7B model, indicating that it is generalizable across models, similar to the Llama circuit.
>
> 	3.  Both Goat and Llama circuits exhibit comparable performance across various models, suggesting that identifying a circuit in one model and applying it to another can yield results as effective as identifying the circuit directly within the target model itself.
>
> | Model      | Full-Model Accuracy | Goat Circuit Accuracy | Llama Circuit Accuracy | Random Circuit Accuracy   | Goat Circuit Faithfulness | Llama Circuit Faithfulness |
> |------------|------------|--------------|---------------|-------------------|--------------|---------------|
> | Llama-7B   |  0.66        |  0.57          |  0.60           |  0.005   |  0.86          |  0.91           |
> | Vicuna-7B  |  0.67        |  0.67          |  0.65           |  0.0342  |  1.00          |  0.96           |
> | Goat-7B    |  0.82        |  0.70          |  0.68           |  0.0044 |  0.85          |  0.83           |
>
>
> -   To understand it thoroughly, we computed the overlap between the attention heads of each of the four groups of heads in both circuits and found that there is a significant overlap between the heads of each of the four groups, as shown in the following table. We observe that more than half of the attention heads are shared between both circuits. We also found that most heads with high causal impact on the final output are shared between both circuits, based on Fig. 6 (Appendix D).
>
> | Head Group(s)     | #Heads in Llama Circuit | #Heads in Goat Circuit | Intersection | Precision   | Recall |
> |------------|------------|--------------|---------------|-------------------|--------------|
> | A   |  40        |  34          |  18           |  0.53    |  0.45          |
> | B  |  9        |  14          |  6           |  0.42  |  0.67          |
> | C   |  22        |  21          |  15           |  0.71  |  0.68          |
> | D   |  4        |  7          |  3           |  0.43  |  0.75          |
> | A + B   |  49        |  48          |  28           |  0.58  |  0.57          |
>
>
> - The findings from the previous table and Fig. 6 suggest that Llama-7B and Goat-7B have roughly the same circuit, with overlapping components that have a high causal impact on the final output. While some of the components are unique, their overall impact is minimal.
>
> -   Please refer to Appendix D for more details.

---

> ### Author Response · Authors · 2023-11-22
> **Response for Reviewer fhvV [2/3]**
>
> **As mentioned in (Wang et al., 2022) [1], faithfulness is not sufficient to prescribe which circuits explain the behavior well. Why do not authors show the completeness and minimality scores, similar to (Wang et al., 2022) [1]?**
>
> **Minimality**
> -   Please note that we do use the minimality criterion from Wang et al. (2022), but instead of reporting the scores we are using it as an optimization criterion; A drawback of path patching is its inability to determine the optimal number of components to incorporate into the circuit. Consequently, there is a high probability that the initially assumed number of components constituting the circuit may contain redundancy. To address this challenge, we employ the minimality criterion to systematically remove redundant attention heads from our initial circuit; unlike Wang et al. (2022), who simply reported the minimality scores without explaining how did they decide on the number of heads to include in the circuit.
>
> -   Specifically, we remove heads that contribute less than 0.5% of the functionality defined by their subcircuit (after removing respective subset K), thus reducing from 100 to 75 attention heads (as described in sec 4.2 in the paper).
>
> -   Further, we are investigating the Llama-7B model which is significantly larger than GPT-2 small. As a result, we had to come up with a greedy approach to compute the subset K for each head in the circuit when computing their minimality score, instead of directly using all the heads in a specific group.
>
> **Completeness**
> -  Following your suggestion, we have added an evaluation with the completeness score from Wang et al. (2022) to Appendix B of the paper. We used the following two sampling methods, proposed in the original paper:
> 	1.  *Random*: We uniformly sample a group of circuit components.
> 	2.  *Circuit subgroups*: We define the subset to be one of four groups of the circuit: A, B, C, and D.
>
> - Results are reported in the following table and Fig. 5 (Appendix B). For the random sampling methods, we report the mean and standard deviation over 20 random subsets.
>
> |           | Full model                      | Circuit                       | Completeness                 |
> |-----------|-----------------------|-----------------------|----------------------|
> | Random         |  0.166 ± 0.028                 |  0.068 ± 0.023                  |  0.098 ± 0.037                 |
> | Group A        |  0.34                            |  0.004                          |  0.336                         |
> | Group B        |  0.152                           |  0.132                          |  0.012                         |
> | Group C        |  0.146                          |  0.144                          |  0.002                         |
> | Group D        |  0.612                         |  0.424                          |  0.188                         |
>
>
> -   When we measure completeness over the circuit, we obtain 0.09, which indicates that, for randomly selected components that have an impact on the task, only about 9% of the performance gap is unexplained by the circuit itself.  For groups B and C, the measurement is even lower, indicating highly complete identification of heads within those groups.  But for groups A and D we obtain 0.34 and 0.19, suggesting a lower level of completeness. Although the performance of the current circuit is quite good, this result suggests that including more heads in groups A and D can improve it further.
>
> -   Please refer to Appendix B for more details.

---

> > ### Author Response · Authors · 2023-11-22
> > **Response for Reviewer fhvV [3/3]**
> >
> > **In Section 4.2: CIRCUIT EVALUATION, the expression F(Cir \ K) - F(Cir \ (K U {v})) / F(Cir \ (K U {v})) in the Minimality paragraph seems wrong. Moreover, the authors mention that they filter out the heads that contribute less than 0.5% of the functionality defined by subset K. This description is inconsistent with the above expression since the denominator of the equation is the functionality defined by the remaining nodes after removing K and v.**
> >
> > -   We acknowledge that there should be an additional parenthesis in the numerator for the concerned expression. We have corrected it in the revised manuscript.
> >
> > -   This description is indeed inconsistent with the above expression. A more accurate description would be that we filter components that do not contribute to any part of the circuit - e.g. contribute less than 0.5% of the functionality defined by the subcircuit (K U {v}), where K is a subgroup that is greedily chosen to emphasize the contribution of v. We have corrected this in the revised version of the paper, thank you for noticing this.
> > ***
> >
> > **What is the motivation to divide heads into four groups (A, B, C, D) instead of three or five? In other words, since authors iteratively identify groups of heads with high direct effects on each other using the path patching score, why did authors end with four groups instead of other numbers?**
> >
> > -   During path patching, we observed that the heads at the correct object token and its next position, which could be influencing heads in Group D, had a minimal effect on the final output. This observation is also supported by our faithfulness measure, where we systematically ablated all the heads except those in the circuit. Despite this, we were able to recover most of the model's performance. Consequently, we decided not to include heads after Group D.
> >
> > -   We applied the path patching algorithm for our iterations for the entity tracking task on Llama-7B. For each iteration, we found a set of heads that either directly or indirectly influenced the final output of the model. Therefore, we divided all the attention heads into four groups A, B, C, and D.

---

### Official Review · Reviewer_M52Y · 2023-11-01

**Soundness:** 3 good
**Presentation:** 3 good
**Contribution:** 3 good
**Rating:** 6
**Confidence:** 4

**Summary:**

The authors investigate the effects of fine-tuning on circuit-level mechanisms within large language models (LLMs) using entity tracking as a focal point. Initially, they employ a path-patching technique to isolate circuits responsible , categorizing attention heads into four groups based on the characteristics inherent to entity tracking. By defining four distinct groups, they ascertain the consistency and effectiveness of these circuits across various models. To delve deeper into the functionality of each group, they outline three main desiderata: Object, Label, and Position. Through activation patching, they discern the role of each group and confirm the commonality of circuit functionality across all models using Desiderata-Based Component Masking. The final phase involves Cross-Model Activation Patching, a method requiring the overlay of activations from similar components of different models on identical inputs. This assists in elucidating how a math-fine-tuned model augments an existing circuit in a base model, leading to enhanced performance in entity tracking. Experiments on LLaMA-7B and its two fine-tuned variants, Vicuna-7B and Goat-7B, reinforce their conclusions.

**Strengths:**

1. The authors effectively elucidate the impact of fine-tuning on the internal computations of large models, particularly with respect to entity tracking. This provides a deeper understanding of how fine-tuning influences model behavior. Utilizing path patching, they constructed four distinct groups of attention heads and enable a granular examination of the model's functionalities. Demonstrated consistency across different models, validating the universality of the identified circuits in performing the entity tracking task.
2. In subsequent tests of the individual capabilities within the identified paths, three out of the four attention head groups exhibited similar functionalities.This substantiates the hypothesis that the circuits in fine-tuned models implement similar functionalities.
3. Experimental results on DCM concerning the Positional Transmitter and Value Fetcher in Goat-7B, as opposed to the original LLaMA-7B, align well with the CAMP experiment. This consistent alignment between the experimental findings and the initial hypotheses strengthens the paper's credibility.

**Weaknesses:**

1. The study's scope is limited to a single foundational model, raising concerns about the generalizability of the conclusions. Without further investigation across a broader spectrum of models, it's challenging to ascertain if the observed mechanisms are universally applicable.
2. When establishing the Desiderata for identifying circuit functionality, the connection between the three tasks and their corresponding abilities remains ambiguously articulated. A clearer exposition of these relationships would have enhanced the clarity and rigor of the study.
3. The section detailing the use of CMAP to patch activations from Goat-7B to Llama-7B lacks clarity, particularly when validating the impact of QKV on the model's performance. The rationale behind why patching the QK-circuit of the Value Fetcher and the value vector of the Position Transmitter heads results in the most significant enhancement is not well-explained.

**Questions:**

see weakness

---

> ### Author Response · Authors · 2023-11-17
> **Response for Reviewer M52Y**
>
> Thanks for the review! We’re glad that you found the paper insightful. We will incorporate your suggestions in an updated version of the paper, but we also wanted to address your comments here:
> ***
>
> **The study's scope is limited to a single foundational model, raising concerns about the generalizability of the conclusions. Without further investigation across a broader spectrum of models, it's challenging to ascertain if the observed mechanisms are universally applicable.**
>
> The methods and workflow employed in this work are generic and can be directly applied to a variety of settings. While we are among the very first to mechanistically study the effect of fine-tuning on LLMs, we have taken care to note in our Limitations section that understanding whether such circuit invariance is universal would require further investigations on additional tasks and models. Our research is intended as a first step, thus focusing on depth rather than breadth, and we believe that our finding that the circuit is invariant across fine-tuned models is surprising, even though observed in a single set of models, and opens the door to further research.
> ***
>
> **When establishing the Desiderata for identifying circuit functionality, the connection between the three tasks and their corresponding abilities remains ambiguously articulated. A clearer exposition of these relationships would have enhanced the clarity and rigor of the study.**
>
> To identify the functionality of various circuit components, we conceptualized and defined three desiderata:
>
>  - *Object desideratum*: It is used to identify circuit components that are encoding the value of correct object information in their output. Consequently, when the output of these components is patched from the counterfactual to the original run, the final output of the original run changes to the correct object of the counterfactual example, as shown in Fig. 2(a).
>  - *Label desideratum*: It is used to identify circuit components that are encoding the query box label value information. Hence, when their output is patched from the counterfactual to the original run, the final output of the original run changes to the object associated with the query box label, of the counterfactual, in the original example, as shown in Fig. 2(b).
>  - *Position desideratum*: It is used to identify circuit components that encode the positional information of the correct object, i.e. when they are patched from counterfactual to the original run, the final output of the original run changes to the object in the original example which is at the same location as the correct object of the counterfactual example, as shown in Fig. 2(c).
>
> For each of the three desiderata, we train a binary mask over the model components to identify the circuit components encoding the corresponding vital information to accomplish the task. *We will elucidate these details in the next version of the manuscript.*
>
> ***
> **The section detailing the use of CMAP to patch activations from Goat-7B to Llama-7B lacks clarity, particularly when validating the impact of QKV on the model's performance. The rationale behind why patching the QK-circuit of the Value Fetcher and the value vector of the Position Transmitter heads results in the most significant enhancement is not well-explained.**
>
> As described in Elhage et. al (2021), the QK (query-key) circuit and OV (output-value) circuit are two independent computations of an attention head. The QK circuit computes the attention pattern and the OV circuit computes the output.
>
> We observe a significant performance gain when the QK-circuit of Value Fetcher heads are patched from Goat-7B to Llama-7B (Fig. 4(b)) which indicates that their capability to attend to the correct object using the positional information forwarded by the Position Transmitter heads has improved during fine-tuning. The computation of the QK-circuit depends on 1) Weight matrices (W_Q & W_K) and 2) their inputs, which in the case of the Value Fetcher heads are primarily constructed from the output of the Position Transmitter heads. Since we observe only a marginal performance increment when patching the output of the Position Transmitter heads across models (Fig. 4(a), purple bar), we attribute the improved capability of Value Fetcher heads to the transformation in the W_Q and W_K matrices of these heads, i.e. fine-tuning has refined these matrices, making them more adept at leveraging positional information to attend to the correct object token. On the other hand, in Position Transmitter heads, the value vector (hence OV-circuit) is responsible (Fig. 4(c)) for the increase of performance, suggesting that their contribution to the performance gain can be attributed to the information encoded in their outputs, which we know from DCM experiments is primarily positional information.
>
> *We will elucidate these details in the next version of the manuscript.*

---

> > ### Comment · Reviewer_M52Y · 2023-11-22
> >
> > Thanks for the authors' response. I'd like to keep my score and look forward to the better version of the paper.

---

### Author Response · Authors · 2023-11-22
**Official Comment by Authors**

We would like to thank each of the reviewers for their thoughtful and constructive feedback. We have responded to each reviewer individually to address their concerns and also updated our paper accordingly. In this summary, we highlight key enhancements that have been implemented, and we encourage additional comments, questions, or suggestions from the reviewers.

-   We have added sections B, D, and E in the Appendix to incorporate supplementary experiments and in-depth details on existing ones.


	-   In section B, we have included the experimental details and results of the completeness evaluation of the circuit identified on Llama-7B.

	-   In section D, we have described the overlap and subtle differences in the circuits identified on Llama-7B and Goat-7B.

	-   In section E, we have elucidated the primary desiderata used to uncover the functionality of circuit components.


-   Following the recommendation from Reviewer fhvV, we have revised our assertion from stating that the exact same circuit is utilized to perform the entity tracking task in fine-tuned models to specifying that a roughly similar circuit is employed. This adjustment has been substantiated by the experimental findings detailed in Section D of the Appendix.

-   Finally, based on the suggestion from Reviewer M52Y, we have also updated section 6.2 to elucidate the results from CMAP experiments.

---

### Meta-Review · Area_Chair_xBV5 · 2023-12-04

**Metareview:**

The paper focuses on entity tracking (model inferring properties with an entity previously defined in the input context) to understand how LMs change during fine-tuning. They use a path-patching technique on synthetic dataset to isolate circuits responsible for entity tracking. They evaluate on LLaMA-7B and their instruction tuned variants, Vicuna-7B and Goat-7B. They find all three models reach high faithfulness scores with the circuit identified in Llama-7B, meaning, models share similar circuit before and after the fine-tuning. Lastly, they introduce cross-model activation patching, which patches activations of the same components of different models, to identify which model parts were responsible for improvements during fine-tuning.

**Justification For Why Not Higher Score:**

The experiments are somewhat limited, only looking at a single LM and its variants.

**Justification For Why Not Lower Score:**

The research question is interesting and provides new insights to important task, even in a simplified setting.

---

### Decision · Program_Chairs · 2024-01-16

Accept (poster)